# MIHC: Multi-View Interpretable Hypergraph Neural Networks with Information Bottleneck for Chip Congestion Prediction

**Zeyue Zhang**[a*]    **Heng Ping**[b*]    **Peiyu Zhang**[b]    **Nikos Kanakaris**[c**]
**Xiaoling Lu**[a]    **Paul Bogdan**[b†]    **Xiongye Xiao**[d†]

[a]Center for Applied Statistics, School of Statistics, Innovation Platform, Renmin University of China;
[b]Ming Hsieh Department of Electrical and Computer Engineering, University of Southern California;
[c]Amazon Web Services, Seattle; and [d]Min H. Kao Department of Electrical Engineering and Computer Science, University of Tennessee, Knoxville

## Abstract

With the advancement of artificial intelligence (AI) and increasing integrated circuit (IC) design complexity, efficient chip design through electronic design automation (EDA) has become critical. Fast and accurate congestion prediction in chip layout and routing can significantly enhance automated design performance. Existing congestion modeling methods are limited by **(i)** ineffective processing and fusion of multi-view circuit data information, and **(ii)** insufficient reliability and interpretability in the prediction process. To address these challenges, we propose the **M**ulti-view **I**nterpretable **H**ypergraph for **C**hip (**MIHC**), a trustworthy multi-view hypergraph neural network framework that **(i)** processes both graph and image information in unified hypergraph representations, capturing topological and geometric circuit data; **(ii)** implements a novel subgraph Information Bottleneck mechanism, identifying critical congestion-correlated regions to guide predictions. This work is the first attempt to incorporate such interpretability into congestion prediction through informative graph reasoning. Experiments show that the MIHC method reduces NMAE by 16.67% and 8.57% in cell-based and grid-based predictions on ISPD2015, and 5.26% and 2.44% on CircuitNet-N28, respectively, compared to state-of-the-art methods. Rigorous cross-design generalization experiments further validate our method's capability to handle entirely unseen circuit designs.

## 1   Introduction

Advances in artificial intelligence (AI) and cyber-physical systems (CPS) [23] require a continuous and increasing demand for computational power supported by integrated circuits (ICs). This necessitates rapid, accurate IC design through electronic design automation (EDA)[12]. Efficient quality assessment of generated circuits is crucial to the design process [9], with circuit congestion prediction being particularly vital as it directly impacts reliability, performance, and manufacturability.

Circuit congestion occurs when routing resources are exceeded by wires attempting to traverse the same physical region [4]. This issue arises when cells (i.e., electronic components) that are logically connected by nets (i.e., connecting wires) are physically placed far apart or when many nets must share limited routing space. These situations commonly result from the mismatch between circuit connectivity requirements and physical layout constraints [10]. Therefore, accurate congestion

---

[*]Equal contribution.

[†]Corresponding authors: pbogdan@usc.edu and xxiao9@utk.edu.

[**]The work does not relate to the author's position at Amazon.

prediction requires comprehensive analysis of both the logical relationships between cells and nets and their geometric distribution in the physical layout [4, 20, 27].

Given this dual nature of congestion formation, circuit congestion prediction primarily involves two types of data: *netlist* and *layout*. *Netlist* captures logical connections between cells and nets, representing topological information naturally modeled as graphs. *Layout* data divides the placement area into grids containing circuit features like cell and net density, encoding geometric information in an image-like format. Effective congestion prediction requires processing and fusion of both complementary information types, as congestion patterns are determined by both connection topology and component geometric distribution.

Traditional congestion prediction approaches use rule-based systems [18] and statistical models [16], but struggle with modern circuit complexity [15]. Deep learning (DL) methods have emerged with superior performance, divided into single-view and multi-view approaches. Single-view methods include DE-HNN [14], which transforms circuit connections into hypergraphs using HGNNs [7] to capture topological information, and GAN-based approaches [26] that process grid-based features as images. However, these single-view approaches inevitably miss valuable information from the complementary perspective.

In an attempt to process both views simultaneously, CircuitGNN [24] incorporates layout information into netlist-based hypergraphs by adding position-aware edges, while Lay-Net[28] introduces specialized HGNN modules into Swin-Transformer-based layout processing. However, these methods tend to favor one view while incorporating the other as supplementary information, leading to insufficient fusion of topological and geometric information. Moreover, existing deep learning approaches operate as black boxes, lacking interpretability and trustworthiness crucial for circuit design validation[9, 12].

To address these limitations, we propose the **M**ulti-view **I**nterpretable **H**ypergraph for **C**hip (MIHC) framework, a trustworthy multi-view hypergraph neural network-based model featuring symmetrical architecture that fuses netlist and layout data through net-based bridging. Specifically, we construct cell-based and grid-based hypergraphs from respective data sources, with hyperedges in both hypergraphs built through nets, enabling balanced fusion of topological and geometric information. Furthermore, we incorporate a novel subgraph information bottleneck (IB) [17] mechanism that identifies critical congestion-correlated regions, guiding focused prediction while revealing areas needing optimization. This enhances accuracy through targeted information processing while providing interpretability by revealing critical local structures causing congestions. Such insights are particularly important since congestion typically emerges from interactions between neighboring components rather than isolated elements, offering designers actionable guidance for targeted optimization.

**Contributions.** The main contributions of this paper are summarized as follows: **(1) Multi-View Representation Learning:** We propose a novel balanced multi-view HGNN that effectively fuses netlist and layout information through net-based bridging, enabling comprehensive learning of both topological and geometric features for congestion prediction. **(2) Interpretable Bottleneck Subgraph:** We develop a subgraph IB mechanism that identifies critical regions highly related to congestion, providing interpretability for circuit optimization while improving prediction accuracy through focused learning. **(3) Superior Prediction Performance:** Comprehensive experiments on the ISPD2015 and CircuitNet-N28 datasets, including rigorous cross-design generalization evaluations, demonstrate that our model reduces NMAE by 16.67% and 8.57% in cell-based and grid-based predictions on ISPD2015, and 5.26% and 2.44% on CircuitNet-N28, respectively, compared to state-of-the-art methods. The evaluation results verify the effectiveness of our multi-view learning strategy and interpretable bottleneck mechanism.

## 2 Related Work

### 2.1 Deep Learning for Congestion Prediction

Congestion prediction involves processing two types of data: netlist data that can be naturally modeled as graphs, and layout data that can be represented in an image format. Based on how these data types are processed, existing DL approaches can be categorized into single-view and multi-view methods.

**Single-View Methods.** These approaches focus on processing either netlist or layout data exclusively. For netlist-based methods, CongestionNet[9] and Cross-Graph[5] model the netlists as homogeneous graphs and employ GNNs for congestion prediction, while DE-HNN[14] leverages HGNNs to capture

high-order relationships in circuit netlists. For layout-based methods, RouteNet[22] treats grid-based layout data as images and utilizes CNNs to generate congestion maps; CircuitFormer[29] regards the grids of a layout as a point cloud and uses Point Transformer to process each grid. However, these single-view approaches inevitably miss valuable information from the other perspective.

**Multi-View Methods.** Recent efforts attempt to incorporate both the netlist and layout information. Netlist-centric approaches like the CircuitGNN[24], LHNN[20], and HybridNet[27] construct hypergraphs from netlists and incorporate layout information through position-aware edges or nodes. Layout-centric methods such as Lay-Net[28], DFM-Net[6], and PGNN[1] primarily process grid-based layout data using vision models like U-Net while integrating the netlist information through specialized GNN or HGNN modules. However, these methods tend to favor one data type, often leading to suboptimal and insufficient fusion of topological and geometric information. In contrast, our proposed MIHC model processes both data types by constructing cell-based and grid-based hypergraphs, achieving balanced fusion through net-based bridging.

## 2.2 Interpretation Methods For Graph

In chip congestion prediction, interpretability has received little attention despite being crucial for identifying key performance-enhancing components and transforming black-box approaches into semi-transparent ones that users find more accessible. Since chip congestion prediction naturally involves graph-structured data with complex interactions, we examine graph domain interpretability methods. Existing approaches are primarily post-hoc, such as GNNExplainer [25] and PGExplainer [13], which provide node/edge-level explanations through learned masks but rely on post-hoc strategies for supervised tasks. Recent progress includes self-interpretable methods like SIGNET [11], which generates inherent interpretations during graph anomaly detection. Inspired by these approaches, our MIHC integrates Information Bottleneck [17] to achieve interpretability in chip congestion prediction—representing, to our knowledge, the first interpretability work in this domain.

# 3 Preliminaries

## 3.1 Problem Formulation

The circuit congestion prediction task aims to estimate the potential congestion levels in a complex circuit design based on its layout and netlist information during early design stages. Congestion can be quantified as the ratio of overflow demand to available routing tracks in IC backend design [3], where overflow represents excess routing requirements beyond provided resources.

Formally, each circuit data is represented as multi-view data comprising grid-based and cell-based information. The grid-based view models circuit layout as a structured grid $P \in \mathbb{R}^{M \times N \times K}$, where $M, N \in \mathbb{N}^+$ are grid dimensions and $K \in \mathbb{N}^+$ is the number of features per grid cell. The cell-based view represents the circuit at a finer granularity level based on the netlist information, where each cell corresponds to an individual circuit component with specific features. The cell-based view represents the circuit netlist as a hypergraph $G = (V, E)$, where $V$ is the set of nodes (cells) and $E$ is the set of hyperedges (nets connecting multiple cells). Given grid-based representation $P$ and cell-based hypergraph $G$, our goal is to simultaneously predict a grid congestion map $C^G \in \mathbb{R}^{M \times N}$ and a cell congestion graph $C^C \in \mathbb{R}^{|V|}$, where $C^G[i, j] \in \mathbb{R}^+$ and $C^C[k] \in \mathbb{R}^+$ quantify congestion levels at grid $(i, j)$ and cell $k$, respectively. The notations used in the manuscript are presented in Appendix B.

The predictive model $f$, parameterized by $\theta$, is defined as follows:

$$f_\theta : (P, G) \mapsto (\hat{C}^G, \hat{C}^C) \tag{1}$$

with the supervised objective to minimize the mean squared error:

$$\mathcal{L}_{\text{sup}} = \frac{1}{M \times N} \sum_{i=1}^{M} \sum_{j=1}^{N} |C^G[i, j] - \hat{C}^G[i, j]|^2 + \frac{1}{|V|} \sum_{k=1}^{|V|} |C^C[k] - \hat{C}^C[k]|^2 \tag{2}$$

## 3.2 Information Bottleneck

The Information Bottleneck (IB) formalism [17] addresses a fundamental challenge in representation learning by extracting the minimal yet sufficient features from high-dimensional data. For example, it

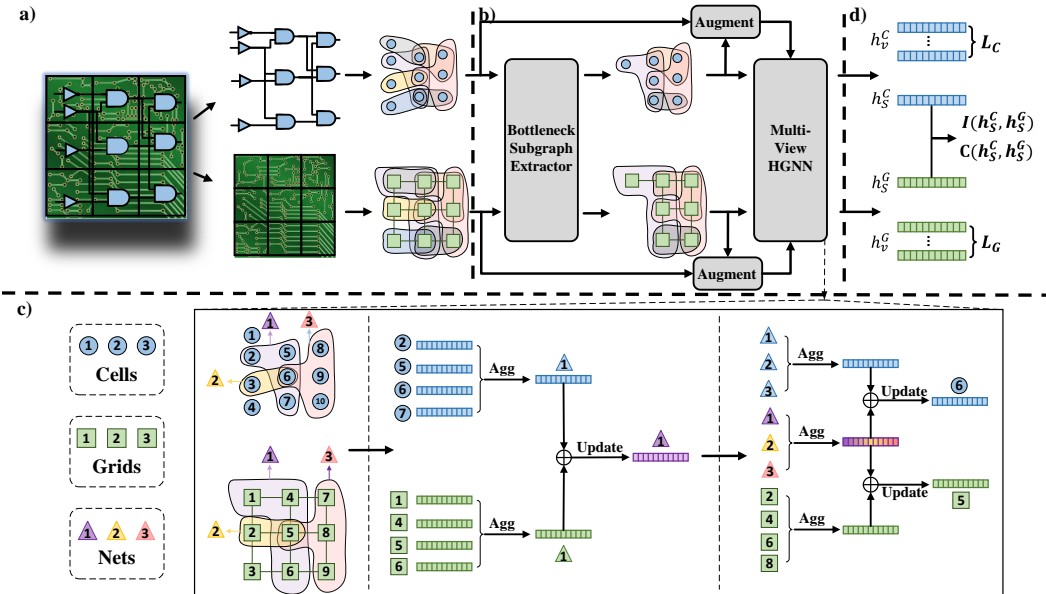

Figure 1: Overview of MIHC framework. (a) Process Module converts raw multi-view data (cell-based netlist and grid-based layout) into hypergraphs $H^C$ and $H^G$ (details in Appendix C.2). (b) Interpretable Subgraph Module applies Multi-View Information Bottleneck to extract congestion-correlated bottleneck subgraphs $H_S^C$ and $H_S^G$. (c) Multi-View Representation Learning Module employs MV-HGNN to capture topological and geometric information, generating node embeddings $h_v^C$, $h_v^G$ and subgraph embeddings $h_S^C$, $h_S^G$. (d) Prediction Module uses MLPs to map embeddings to congestion predictions, optimized through supervised, IB, and contrastive losses. ($\mathcal{L}_{\text{sup}}$, $\mathcal{L}_{\text{IB}}$, $\mathcal{L}_{\text{cont}}$).

inspired new hierarchical perception methods for multimodal learning [21]. Given an input variable $X$ and target variable $Y$, IB seeks an intermediate representation $T$ that optimally balances data compression and target information preservation, formulated as:

$$\mathcal{L}[p(t|x)] = I(T;X) - \beta^* I(T;Y), \tag{3}$$

where $I(T;X)$ and $I(T;Y)$ denote mutual information, and $\beta^* \in \mathbb{R}^+$ controls the compression-relevance trade-off. This formulation enables the identification of critical features driving model predictions while minimizing redundancy, making IB particularly valuable for model interpretability and subgraph analysis in networks [11].

## 4 Method

In this section, we introduce the proposed MIHC framework (see Figure 1) which consists of four parts: (a) The *Process Module* converts raw multi-view data, including cell-based netlist graph and grid-based layout image, into hypergraphs for further processing. (b) The *Interpretable Subgraph Module* introduces a novel subgraph Information Bottleneck (IB) mechanism to identify critical bottleneck subgraphs highly related to circuit congestion and guide the model to focus on these regions during prediction. (c) The *Multi-View Representation Learning Module* integrates both graph and image data into unified hypergraph representations, capturing topological and geometric features of circuit data. (d) The *Prediction Module* computes the prediction in both cell and grid views.

### 4.1 Process Module

Given the dual nature of the data, the Process Module converts raw multi-view data, including both cell-based netlist graph and grid-based layout image, into hypergraphs for subsequent processing.

**Hypergraph Construction from Cell-based Netlist Graph.** We construct a cell-based hypergraph $H^C = (V^C, E^C)$, where $V^C$ represents circuit cells and $E^C$ represents hyperedges based on net

connections. Each node $v^C \in V^C$ is initialized with feature $X_v^C$ derived from circuit netlist characteristics. This hypergraph primarily captures topological relationships among circuit components.

**Hypergraph Construction from Grid-based Layout Image.** We construct a grid-based hypergraph $H^G = (V^G, E^G)$, where $V^G$ represents local layout image regions with feature representation $X_v^G$ containing each region's component density, and $E^G$ represents hyperedges connecting these regions. The formation of grid-based hyperedges $e^G \in E^G$ is guided by spatial proximity and net connections. Hyperedges are established among nodes corresponding to spatially adjacent regions to preserve the geometric information. More importantly, nodes within the same net are grouped into hyperedges to capture high-order geometric information. Both cell-based hyperedges $E^C$ and grid-based hyperedges $E^G$, though from different hypergraphs, are built on identical net connections—critical for fusing topological and geometric information in subsequent multi-view representation learning.

## 4.2 Interpretable Subgraph Module

The interpretable subgraph module is designed to identify critical parts in the circuit hypergraph and guide the model to assign more attention to these parts.

**Multi-View Information Bottleneck.** We develop a Multi-View Information Bottleneck (MVIB) approach combining information from both cell-based netlist and grid-based layout data. After extracting hypergraphs $H^C$ and $H^G$, IB learns bottleneck subgraphs for each view. Taking $H^C$ as an example, the objective is to learn a bottleneck subgraph $H_S^C$ by optimizing the following objective:

$$\max_{H_S^C} I(H^G; H_S^C) - \beta^C I(H^C; H_S^C \mid H^G), \tag{4}$$

where $\beta^C$ controls compression-relevance trade-off. A similar formulation applies for $H_S^G$.

**Bottleneck Subgraph Extractor.** Inspired by [11], we adopt a single extractor design for both hypergraph views. The extractor processes hypergraph $H^C$ using a Hypergraph Neural Network to obtain node probabilities, modeled as $p_\theta(H_S^C|H^C) = \prod_{v \in V^C} p_\theta(v \in V_S^C|H^C)$. The HNN outputs probability vector $\mathbf{p} \in [0,1]^{|V^C|}$, enabling bottleneck subgraph extraction:

$$H_S^C = (\mathbf{M}^C, \mathbf{X}_S^C) = (\mathbf{M}^C, \mathbf{X}^C \odot \mathbf{p}), \tag{5}$$

where $\mathbf{M}^C$ is the incidence matrix of $H^C$, $\mathbf{X}^C$ represents the node features, and $\odot$ denotes the row-wise production. The bottleneck subgraph for the grid-based view $H_S^G$ can then be extracted using the mapped probability vector in the same form as Eq. (5).

Node probabilities map between views through spatial relationships. For each node $v^G$ in $H^G$, its probability is $\mathbf{p}^*[v^G] = \max_{v^C \in \mathcal{N}(v^G)} \mathbf{p}[v^C]$, where $\mathcal{N}(v^G)$ denotes cell nodes associated with grid node $v^G$. This single-extractor design ensures bottleneck subgraph consistency across views while reducing complexity. The extracted subgraphs $H_S^C$ and $H_S^G$ preserve the most informative structures for congestion prediction, guided by the information bottleneck objective in Eq. (7).

**Bottleneck Subgraph Augmentation.** After extracting the bottleneck subgraphs, we leverage bottleneck subgraphs to enhance feature representations through dynamic reweighting:

$$H^C = (\mathbf{M}^C, \mathbf{X}^C \cdot (1 + \sigma(W^C[\mathbf{X}^C \| \mathbf{X}_S^C]))), \quad H^G = (\mathbf{M}^G, \mathbf{X}^G \cdot (1 + \sigma(W^G[\mathbf{X}^G \| \mathbf{X}_S^G]))), \tag{6}$$

where $W^C$, $W^G$ are learnable weights, $\sigma$ is the sigmoid function, and $\|$ represents concatenation. This mechanism enhances features based on bottleneck structure relevance, while residual connections ensure stable gradient flow and preserve structural information.

**Multi-View Information Bottleneck Loss.** The information bottleneck loss regularizes bottleneck subgraph embeddings. The objective is reformulated as:

$$\mathcal{L}_{MVIB} = -I(H_S^C; H_S^G) + \beta D_{SKL} \left( p_\theta(H_S^C \mid H^C) \| p_\gamma(H_S^G \mid H^G) \right), \tag{7}$$

where $p_\theta$ and $p_\gamma$ refer to bottleneck subgraph extractors, $D_{SKL}$ denotes symmetrized KL divergence, and $\beta$ controls the trade-off between terms. Derivations from Eq. (4) Eq.(7) are in Appendix A.

## 4.3 Multi-View Representation Learning Module

Given the multi-view data available, we propose a Multi-View Heterogeneous Graph Neural Network (MV-HGNN) that processes and integrates cell-based and grid-based hypergraph representations through a two-step message passing mechanism.

**Node-to-Hyperedge Message Passing.** For both cell-based and grid-based hypergraphs, we first aggregate node features within each hyperedge separately:

$$h_e^C = \sum_{v \in \mathcal{V}^\mathcal{C}(e)} W_v^C h_v^C, \quad h_e^G = \sum_{v \in \mathcal{V}^\mathcal{G}(e)} W_v^G h_v^G, \tag{8}$$

where $h_e^C$, $h_e^G$ are intermediate hyperedge embeddings, $\mathcal{V}^\mathcal{C}(e)$, $\mathcal{V}^\mathcal{G}(e)$ denote nodes in respective hyperedges, and $W_v^C$, $W_v^G$ can transform features into a common semantic space.

The net-based hyperedge embedding is then obtained by first concatenating these intermediate embeddings followed by a non-linear transformation:

$$h_e^N = \sigma(W_e^N[h_e^C || h_e^G]), \tag{9}$$

where $h_e^N$ represents the actual net-based hyperedge embedding that captures both topological and geometric information.

**Hyperedge-to-Node Message Passing.** The second step updates node embeddings by aggregating information from different sources. For cell-based nodes:

$$\hat{h}_v^C = \sum_{e \in \mathcal{E}^C(v)} W_e^C h_e^C, \quad \hat{h}_v^N = \sum_{e \in \mathcal{E}^N(v)} W_e^N h_e^N, \tag{10}$$

where $\hat{h}_v^C$, $\hat{h}_v^N$ are intermediate embeddings from respective hyperedges, $\mathcal{E}^C(v)$, $\mathcal{E}^N(v)$ denote hyperedge sets containing node $v$. The final cell-based node embedding is obtained through concatenation and non-linear transformation:

$$h_v^{C'} = \sigma(W_v^{C'}[\hat{h}_v^C || \hat{h}_v^N]). \tag{11}$$

For grid-based nodes, we similarly aggregate information from spatially adjacent nodes and use the previously computed net-based hyperedge information: $\hat{h}_v^G = \sum_{u \in \mathcal{N}^\mathcal{G}(v)} W_u^G h_u^G$, where $\mathcal{N}^\mathcal{G}(v)$ represents spatially adjacent grid nodes. The final grid-based embedding:

$$h_v^{G'} = \sigma(W_v^{G'}[\hat{h}_v^G || \hat{h}_v^N]). \tag{12}$$

After $L$ message passing layers, final node embeddings serve two purposes: downstream congestion prediction and generating subgraph embeddings for contrastive learning. The subgraph embeddings are derived from previously identified bottleneck subgraphs:

$$h_S^C = \text{Aggregate}(\{h_v^{C'} \mid v \in V_S^C\}), \quad h_S^G = \text{Aggregate}(\{h_v^{G'} \mid v \in V_S^G\}). \tag{13}$$

This symmetric design facilitates effective congestion prediction for both views while maintaining the inherent characteristics of each representation.

**Contrastive Loss.** To enhance and align the interaction between cell-based and grid-based information, we employ contrastive learning. The contrastive loss is defined as:

$$\mathcal{L}_{\text{cont}} = -\sum_{i=1}^{N} \log \frac{\exp(\text{sim}(h_{S_i}^C, h_{S_i}^G)/\tau)}{\sum_{j=1}^{N} \exp(\text{sim}(h_{S_i}^C, h_{S_j}^G)/\tau)}, \tag{14}$$

where $h_{S_i}^C$, $h_{S_i}^G$ form positive pairs from the same input, $h_{S_j}^G$ represents negative samples, $\text{sim}(a,b)$ denotes cosine similarity, and $\tau$ controls similarity distribution sharpness.

### 4.4 Prediction Module

Unlike previous methods that can only produce predictions in either the cell view or the grid view at once, our method's prediction module can output predictions in both scenarios simultaneously.

**Prediction.** In the prediction, we obtain both $h_v^C$, the cell-based node embedding, and $h_v^G$, the grid-based node embedding. Therefore, we use two MLPs, $f^C$ and $f^G$, to map the two embeddings to the node congestion prediction results for both views: $\hat{y}_v^C = f^C(h_v^C), \quad \hat{y}_v^G = f^G(h_v^G)$, where $\hat{y}_v^C$ and $\hat{y}_v^G$ represent the predicted node congestion values for the cell and grid views, respectively.

**Total Loss.** The overall loss combines contrastive (refer to Eq.(14)), supervised (refer to Eq.(2)), and information bottleneck losses (refer to Eq.(7)) to optimize the model.

$$\mathcal{L}_{\text{total}} = \mathcal{L}_{\text{cont}} + \mathcal{L}_{\text{IB}} + \mathcal{L}_{\text{sup}}. \tag{15}$$

**Time Complexity.** The overall complexity is $O((|V_C| \times |E_C| + |V_G| \times |E_G|) \times D \times L)$, where $|V_C|$ and $|V_G|$ represent the number of cell and grid nodes, $|E_C|$ and $|E_G|$ represent their respective hyperedges, $D$ is the hidden dimension, and $L$ is the number of layers.

Table 1: Placement-level results on All Datasets for both Cell- and Grid-based View. The best in **bold** and the second best underlined.

| View | Model | ISPD2015-B | | | | | ISPD2015-F | | | | | CircuitNet-N28 | | | | |
|---|---|---|---|---|---|---|---|---|---|---|---|---|---|---|---|---|
| | | NMAE↓ | NRMS↓ | Pearson↑ | Spearman↑ | Kendall↑ | NMAE↓ | NRMS↓ | Pearson↑ | Spearman↑ | Kendall↑ | NMAE↓ | NRMS↓ | Pearson↑ | Spearman↑ | Kendall↑ |
| Cell-based | GCN | 0.038 | 0.044 | 0.547 | 0.498 | 0.421 | 0.051 | 0.058 | 0.344 | 0.336 | 0.309 | 0.045 | 0.051 | 0.516 | 0.438 | 0.319 |
| | GAT | 0.039 | 0.045 | 0.531 | 0.517 | 0.463 | 0.050 | 0.057 | 0.351 | 0.342 | 0.298 | 0.046 | 0.051 | 0.513 | 0.447 | 0.321 |
| | HGNN | 0.036 | 0.042 | 0.577 | 0.553 | 0.476 | 0.045 | 0.052 | 0.410 | 0.366 | 0.309 | 0.044 | 0.048 | 0.552 | 0.468 | 0.332 |
| | CircuitGNN | 0.034 | 0.040 | 0.598 | 0.611 | 0.487 | 0.046 | 0.053 | 0.402 | 0.378 | 0.316 | 0.040 | 0.043 | 0.609 | 0.500 | 0.368 |
| | LHNN | 0.033 | 0.038 | 0.629 | 0.627 | 0.502 | 0.043 | 0.049 | 0.448 | 0.384 | 0.364 | 0.039 | 0.043 | 0.617 | 0.509 | 0.366 |
| | DE-HNN | 0.032 | 0.037 | 0.645 | 0.632 | 0.520 | 0.042 | 0.048 | 0.467 | 0.395 | 0.381 | 0.038 | 0.042 | 0.638 | 0.511 | 0.374 |
| | **MIHC (Ours)** | **0.029** | **0.034** | **0.687** | **0.689** | **0.574** | **0.035** | **0.041** | **0.524** | **0.512** | **0.447** | **0.036** | **0.039** | **0.661** | **0.522** | **0.396** |
| Grid-based | GCN | 0.043 | 0.049 | 0.405 | 0.162 | 0.137 | 0.046 | 0.053 | 0.357 | 0.152 | 0.131 | 0.052 | 0.053 | 0.417 | 0.399 | 0.312 |
| | GAT | 0.044 | 0.050 | 0.398 | 0.144 | 0.141 | 0.048 | 0.055 | 0.331 | 0.137 | 0.116 | 0.052 | 0.054 | 0.411 | 0.405 | 0.323 |
| | HGNN | 0.040 | 0.046 | 0.451 | 0.184 | 0.142 | 0.043 | 0.050 | 0.352 | 0.147 | 0.121 | 0.051 | 0.052 | 0.442 | 0.424 | 0.331 |
| | RouteNet | 0.038 | 0.045 | 0.522 | 0.198 | 0.157 | 0.042 | 0.050 | 0.362 | 0.161 | 0.137 | 0.047 | 0.049 | 0.541 | 0.501 | 0.373 |
| | CircuitGNN | 0.037 | 0.043 | 0.561 | 0.201 | 0.164 | 0.042 | 0.049 | 0.364 | 0.162 | 0.134 | 0.047 | 0.048 | 0.547 | 0.508 | 0.371 |
| | CircuitFormer | 0.034 | 0.040 | 0.649 | 0.231 | 0.169 | 0.038 | 0.045 | 0.453 | 0.196 | 0.165 | 0.044 | 0.046 | 0.622 | 0.526 | 0.384 |
| | LHNN | 0.033 | 0.040 | 0.651 | 0.252 | 0.217 | 0.037 | 0.044 | 0.464 | 0.221 | 0.182 | 0.043 | 0.044 | 0.646 | 0.533 | 0.389 |
| | Lay-Net | 0.031 | 0.037 | 0.667 | **0.301** | 0.248 | 0.035 | 0.042 | 0.484 | 0.256 | 0.208 | 0.041 | 0.043 | 0.671 | 0.558 | **0.406** |
| | **MIHC (Ours)** | **0.030** | **0.036** | **0.675** | 0.297 | **0.252** | **0.032** | **0.039** | **0.503** | **0.271** | **0.227** | **0.040** | **0.041** | **0.682** | **0.572** | 0.402 |

Table 2: Design-level results on ISPD2015-B and ISPD2015-F for both Cell- and Grid-based View. The best in **bold** and the second best underlined.

| Direction | View | Model | ISPD2015-B | | | | | ISPD2015-F | | | | |
|---|---|---|---|---|---|---|---|---|---|---|---|---|
| | | | NMAE↓ | NRMS↓ | Pearson↑ | Spearman↑ | Kendall↑ | NMAE↓ | NRMS↓ | Pearson↑ | Spearman↑ | Kendall↑ |
| A→B | Cell-based | CircuitGNN | 0.058 | 0.064 | 0.376 | 0.335 | 0.248 | 0.071 | 0.077 | 0.343 | 0.304 | 0.218 |
| | | LHNN | 0.054 | 0.060 | 0.398 | 0.355 | 0.265 | 0.067 | 0.073 | 0.368 | 0.321 | 0.235 |
| | | DE-HNN | 0.051 | 0.057 | 0.415 | 0.372 | 0.282 | 0.063 | 0.069 | 0.385 | 0.342 | 0.252 |
| | | **MIHC (Ours)** | **0.048** | **0.053** | **0.448** | **0.405** | **0.302** | **0.059** | **0.065** | **0.414** | **0.375** | **0.277** |
| | Grid-based | CircuitGNN | 0.061 | 0.067 | 0.366 | 0.158 | 0.132 | 0.074 | 0.080 | 0.331 | 0.148 | 0.122 |
| | | LHNN | 0.057 | 0.063 | 0.389 | 0.172 | 0.145 | 0.070 | 0.076 | 0.349 | 0.162 | 0.135 |
| | | Lay-Net | 0.053 | 0.059 | 0.411 | 0.192 | 0.165 | 0.064 | 0.070 | 0.394 | 0.200 | **0.174** |
| | | **MIHC (Ours)** | **0.051** | **0.057** | **0.436** | **0.218** | **0.188** | **0.064** | **0.069** | **0.405** | **0.201** | 0.167 |
| B→A | Cell-based | CircuitGNN | 0.063 | 0.068 | 0.351 | 0.325 | 0.232 | 0.075 | 0.081 | 0.337 | 0.293 | 0.205 |
| | | LHNN | 0.058 | 0.065 | 0.379 | 0.347 | 0.246 | 0.070 | 0.076 | 0.354 | 0.315 | 0.222 |
| | | DE-HNN | 0.053 | 0.058 | 0.405 | 0.365 | 0.269 | 0.065 | 0.071 | 0.378 | 0.336 | 0.238 |
| | | **MIHC (Ours)** | **0.049** | **0.055** | **0.428** | **0.384** | **0.281** | **0.062** | **0.067** | **0.399** | **0.357** | **0.262** |
| | Grid-based | CircuitGNN | 0.064 | 0.070 | 0.348 | 0.148 | 0.122 | 0.077 | 0.083 | 0.318 | 0.138 | 0.112 |
| | | LHNN | 0.060 | 0.066 | 0.369 | 0.162 | 0.135 | 0.073 | 0.079 | 0.337 | 0.152 | 0.125 |
| | | Lay-Net | 0.055 | 0.061 | 0.392 | 0.197 | **0.179** | 0.068 | 0.073 | 0.367 | **0.191** | 0.152 |
| | | **MIHC (Ours)** | **0.054** | **0.059** | **0.415** | **0.202** | 0.175 | **0.067** | **0.072** | **0.388** | 0.182 | **0.168** |

# 5 Experiments

In this section, we evaluate MIHC through comprehensive experiments designed to address four key research questions:

**RQ1:** How does MIHC's circuit congestion prediction accuracy compare to state-of-the-art methods?

**RQ2:** What is the MIHC's interpretability performance?

**RQ3:** What is the impact of MIHC's core design components on its overall performance?

**RQ4:** How sensitive is MIHC's performance to parameter variations?

## 5.1 Experimental Setup

**Datasets.** For chip congestion prediction, we evaluate our method on two public datasets: ISPD2015 [2] and CircuitNet-N28 [3]. ISPD2015 is divided into two variants: ISPD2015-Balanced (ISPD2015-B), which excludes extremely large superblue circuits, and ISPD2015-Full (ISPD2015-F), which includes them, allowing us to assess our model's performance on data imbalance challenges. CircuitNet-N28 contains over 10,000 samples (compared to ISPD2015's approximately 500), providing a more comprehensive test of our model's generalizability across diverse circuit designs.

To rigorously evaluate generalization capability, we employ two data partitioning strategies: (1) Placement-level split, where different placement solutions from the same design appear in both training and test sets, and (2) Design-level split, where training and test sets contain completely different circuit designs with no overlap. The placement-level split follows our standard 7:3 train-test ratio, while the design-level split involves bidirectional evaluation (A→B and B→A) on carefully partitioned design groups. Detailed partitioning strategies are presented in Appendix C.5.

For explainable GAD task, we use MNIST0 and MNIST1 datasets [11].

**Baselines.** There are two aspects of prediction tasks: cell-based prediction and grid-based prediction. To comprehensively evaluate our model, we first employ several classical graph learning models as

baselines: GCN [8], GAT [19], and HGNN [7]. Furthermore, we compare with specialized circuit-oriented models: CircuitGNN [24] and LHNN [20], which are capable of handling both cell-based and grid-based predictions; DE-HNN [14], which focuses exclusively on cell-based prediction; and RouteNet [22], CircuitFormer [29] and Lay-Net [28], which specifically target grid-based prediction. To illustrate the usefulness of the model interpretability module, we chose GCN + GNNExplainer [25], GAT + GNNExplainer and SIGNET [11] as baselines.

**Evaluation Metrics.** For chip congestion prediction, similar to [24], we employ both regression-based metrics (NMAE and NRMS) to measure prediction accuracy and ranking-based metrics (Pearson, Spearman, and Kendall correlations) to assess prediction quality; while in interpretability module test, since the downstream task is a classification task, ACC, Precision, Recall, F1, and AUC are used to measure the accuracy of the classification.

Further details about datasets and our experiment settings are presented in Appendix C.

## 5.2 Chip Congestion Prediction Results (RQ1)

We present comprehensive evaluation results under both placement-level and design-level settings. The placement-level results are shown in Table 1 and design-level results are presented in Table 2, both with the best in **bold** and the second best underlined (More details in Appendix D.1). Given that our data encompasses both cell and grid views, we assess the validity and rationality of our model across these dual views by comparing prediction accuracy against their respective state-of-the-art (SOTA) counterparts.

### 5.2.1 Placement-level Results

As shown in Table 1, for ISPD2015-B, MIHC outperforms the SOTA method DE-HNN with 9.37% reduction in NMAE and 6.51% improvement in Pearson correlation at the cell level, and surpasses Lay-Net with 3.23% reduction in NMAE and 1.20% improvement in Pearson correlation at the grid level. On the more challenging ISPD2015-F with extremely large circuits, MIHC achieves even more significant gains: 16.67% reduction in NMAE and 12.21% improvement in Pearson correlation compared to DE-HNN at the cell level, and 8.57% reduction in NMAE and 3.93% improvement in Pearson correlation over Lay-Net at the grid level. These results demonstrate MIHC's robustness when handling imbalanced data distributions. The CircuitNet-N28 results, based on over 10,000 diverse samples, further confirm MIHC's generalizability with 5.26% NMAE reduction and 3.60% Pearson improvement compared to DE-HNN at the cell level, and 2.44% NMAE reduction and 1.64% Pearson improvement over Lay-Net at the grid level. This consistent performance across a substantially larger dataset validates our method's universal applicability for congestion prediction in real-world VLSI design environments.

### 5.2.2 Design-level Results

To rigorously evaluate generalization to entirely unseen circuit designs, we conduct additional experiments following the design-level split setting (detailed setup in Appendix C.5), where training and test sets contain completely different circuit designs with no overlap. As shown in Table 2, this setting presents significantly more challenging conditions with substantial performance degradation across all methods. Despite these challenges, MIHC maintains superior performance over baselines in the vast majority of metrics. For ISPD2015-B, MIHC outperforms DE-HNN with 7.55% reduction in cell-based NMAE and 5.68% improvement in Pearson correlation in the B→A experiments, while achieving 5.87% Pearson improvement in grid-based prediction over Lay-Net. For ISPD2015-F, MIHC achieves 6.35% cell-based NMAE reduction and 7.53% Pearson improvement in the A→B direction. These results provide strong evidence that our method achieves genuine generalization capability rather than simply memorizing design-specific characteristics, offering a realistic evaluation for real-world deployment scenarios.

## 5.3 Explainability Results (RQ2)

To validate the effectiveness of our model's explainability module, we examine its capability to identify important structures in chip congestion prediction scenarios. As demonstrated in Figure 2, our bottleneck subgraph successfully captures critical congestion regions, and MIHC achieves

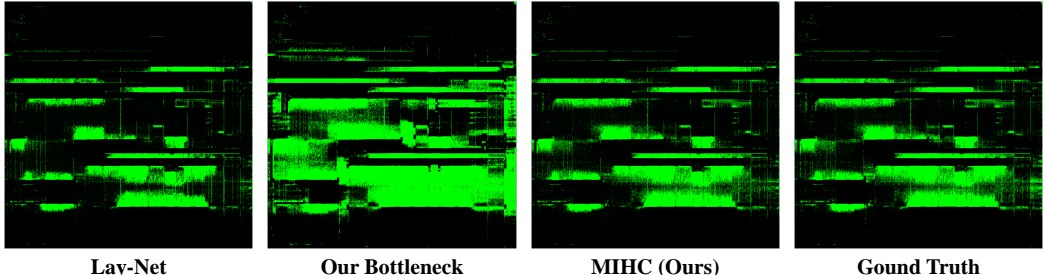

| **Lay-Net** | **Our Bottleneck** | **MIHC (Ours)** | **Gound Truth** |

Figure 2: Visualization of heat maps on ISPD2015/mgc_superblue16_a design showing Lay-Net, our bottleneck subgraph, MIHC (Ours), and ground truth congestion maps.

Table 3: Ablation Study of Model Modules on ISPD2015-B. The best in **bold**.

| Model Variants | Cell-based Results | | | | | Grid-based Results | | | | |
|---|---|---|---|---|---|---|---|---|---|---|
| | NMAE↓ | NRMS↓ | Pearson↑ | Spearman↑ | Kendall↑ | NMAE↓ | NRMS↓ | Pearson↑ | Spearman↑ | Kendall↑ |
| HGNN | 0.036 | 0.042 | 0.577 | 0.553 | 0.476 | 0.040 | 0.046 | 0.451 | 0.184 | 0.142 |
| w/o MV-HGNN | 0.036 (0.0%) | 0.041 (+2.4%) | 0.580 (+0.5%) | 0.571 (+3.3%) | 0.477 (+0.2%) | 0.039 (+2.5%) | 0.044 (+4.3%) | 0.482 (+6.9%) | 0.187 (+1.6%) | 0.144 (+1.4%) |
| w/o ISM | 0.031 (+13.9%) | 0.036 (+14.3%) | 0.654 (+13.3%) | 0.643 (+16.3%) | 0.536 (+12.6%) | 0.033 (+17.5%) | 0.039 (+15.2%) | 0.652 (+44.6%) | 0.263 (+42.9%) | 0.231 (+62.7%) |
| Full Model | **0.029 (+19.4%)** | **0.034 (+19.0%)** | **0.687 (+19.1%)** | **0.689 (+24.6%)** | **0.574 (+20.6%)** | **0.030 (+25.0%)** | **0.036 (+21.7%)** | **0.675 (+49.7%)** | **0.297 (+61.4%)** | **0.252 (+77.5%)** |

Table 4: Ablation Study of Different Loss Functions on ISPD2015-B.

| Loss Components | | | | Cell-based Results | | | | | Grid-based Results | | | | |
|---|---|---|---|---|---|---|---|---|---|---|---|---|---|
| Cell | Grid | IB | Cont | NMAE↓ | NRMS↓ | Pearson↑ | Spearman↑ | Kendall↑ | NMAE↓ | NRMS↓ | Pearson↑ | Spearman↑ | Kendall↑ |
| ✓ | ✗ | ✗ | ✗ | 0.033 | 0.038 | 0.639 | 0.622 | 0.511 | 0.037 | 0.042 | 0.612 | 0.222 | 0.205 |
| ✗ | ✓ | ✗ | ✗ | 0.035 (-6.06%) | 0.040 (-5.26%) | 0.596 (-6.73%) | 0.605 (-2.73%) | 0.484 (-5.28%) | 0.034 (+8.11%) | 0.040 (+4.76%) | 0.650 (+6.21%) | 0.254 (+14.41%) | 0.241 (+17.56%) |
| ✓ | ✓ | ✗ | ✗ | 0.031 (+6.06%) | 0.036 (+5.26%) | 0.659 (+3.13%) | 0.658 (+5.79%) | 0.550 (+7.63%) | 0.033 (+10.81%) | 0.039 (+7.14%) | 0.661 (+8.01%) | 0.264 (+18.92%) | 0.243 (+18.54%) |
| ✓ | ✓ | ✗ | ✓ | 0.030 (+9.09%) | 0.036 (+5.26%) | 0.662 (+3.60%) | 0.660 (+6.11%) | 0.554 (+8.41%) | 0.033 (+10.81%) | 0.038 (+9.52%) | 0.666 (+8.82%) | 0.269 (+21.17%) | 0.246 (+20.00%) |
| ✓ | ✓ | ✓ | ✗ | **0.029 (+12.12%)** | 0.035 (+7.89%) | 0.672 (+5.16%) | 0.665 (+6.91%) | 0.559 (+9.39%) | 0.031 (+16.22%) | 0.038 (+9.52%) | 0.671 (+9.64%) | 0.284 (+27.93%) | 0.249 (+21.46%) |
| ✓ | ✓ | ✓ | ✓ | **0.029 (+12.12%)** | **0.034 (+10.53%)** | **0.687 (+7.51%)** | **0.689 (+10.77%)** | **0.574 (+12.33%)** | **0.030 (+18.92%)** | **0.036 (+14.29%)** | **0.675 (+10.29%)** | **0.297 (+33.78%)** | **0.252 (+22.93%)** |

predictions that closely match the ground truth patterns, underscoring its superior interpretability. Additionally, the bottleneck subgraph heat map indicates that our IB bottleneck subgraph can be extracted to reveal regions relevant to the final prediction results. While node-level predictions identify congestion points, our Information Bottleneck identifies the critical local structures that cause these anomalies, revealing the structural patterns that contribute to surrounding node congestion rather than just detecting generic anomalies. This is particularly important in circuit design where congestion typically emerges from interactions between neighboring components rather than isolated elements. Due to the lack of publicly available datasets for interpretability in the EDA domain, we also briefly evaluated our model on the MNIST 0/1 dataset, where it demonstrated strong consistency across various metrics in Table 16. In comparison, baseline methods showed significant limitations: GAT+GNNExplainer suffered from low precision while SIGNET failed to capture comprehensive node relationships. These results, combined with the visual evidence from chip congestion heatmaps, confirm that MIHC not only delivers superior prediction accuracy but also provides meaningful explainability for practical VLSI design applications.

## 5.4 Ablation Study(RQ3)

In ablation studies, we conducted two sets of experiments on ISPD2015-B to evaluate: the contribution of different model modules (Table 3) and the impact of different loss functions (Table 4).

The ablation results of model modules in Table 3 reveal several important findings. First, removing the Multi-View Representation Learning Module (w/o MV-HGNN) results in performance similar to the basic HGNN but significantly worse than the Full Model, demonstrating the crucial role of multi-view information fusion. Second, excluding the Interpretable Subgraph Module (w/o ISM) yields better results than basic HGNN but still underperforms compared to the Full Model, with substantial decreases in correlation metrics (13.3% lower Pearson for cell-based results). Finally, the Full Model achieves optimal performance across all metrics by effectively combining ISM for critical information filtering with MV-HGNN for multi-view integration, showing impressive improvements over the baseline HGNN (19.1% Pearson improvement for cell-based results).

The comprehensive ablation results in Table 4 systematically demonstrate the effectiveness evaluation of different loss functions. The experimental results show that using both Cell and Grid Supervised loss consistently achieves better performance (NMAE=0.031 for cell-based, 0.033 for grid-based)

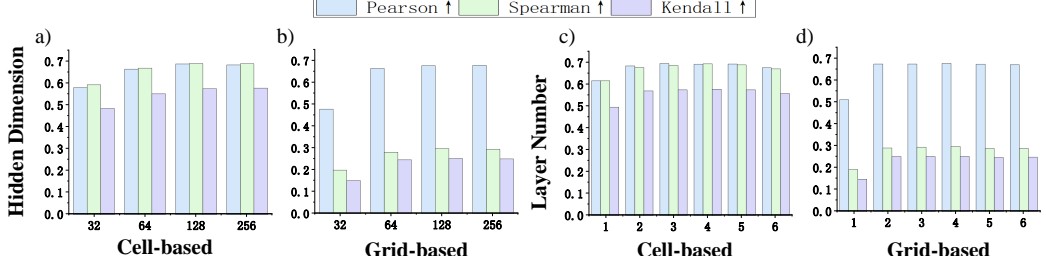

Figure 3: Hyperparameter sensitivity analysis: Impact of hidden dimensions (a,b) and MV-HGNN layer number (c,d) on cell-based and grid-based performance.

compared to single-view supervision (Cell-Sup only: 0.033, 0.037; Grid-Sup only: 0.035, 0.034), thus validating the fundamental importance of multi-view information fusion by using multi-view loss. Furthermore, through detailed comparative analysis, the removal of IB loss (w/o IB Loss) leads to substantial performance degradation (Pearson correlation decreases by 4.21% for cell-based results), confirming its crucial role in guiding the model to extract essential subgraph components.

## 5.5 Parameters Sensitivity(RQ4)

We conduct a comprehensive hyperparameter analysis of MIHC on ISPD2015-B, specifically examining the impact of Multi-View HGNN layer number (ranging from 1 to 6) and hidden dimensions (varying from 32 to 256) on model performance. As illustrated in Figure 3, MIHC demonstrates robust stability across different hyperparameter configurations. However, notable performance degradation occurs in two specific configurations: when the model is configured with a single Multi-View HGNN layer or when the hidden dimension is set to 32. These significant performance drops can be attributed to insufficient model capacity, as either a single layer or a small hidden dimension proves inadequate for learning the complex hierarchical features and relationships inherent in the data.

## 6 Conclusion

This paper presents MIHC, a novel interpretable hypergraph neural network for circuit congestion prediction. Through its balanced multi-view hypergraph architecture with net-based bridging, MIHC achieves effective fusion of netlist and layout information, providing a comprehensive understanding of circuit characteristics. The incorporation of a subgraph Information Bottleneck mechanism further enhances the model by identifying critical subgraphs correlated with congestion patterns and leveraging this selective information to improve prediction accuracy. Our comprehensive experiments demonstrate MIHC's superior performance compared to existing methods, achieving significant improvements in both cell-based and grid-based prediction tasks. This synergistic combination of balanced multi-view information fusion and interpretable bottleneck analysis represents a meaningful advancement in trustworthy Electronic Design Automation, establishing a foundation for future interpretable machine learning approaches in integrated circuit design.

**Limitations.** While MIHC achieves significant performance improvements, there remain areas for enhancement. The Information Bottleneck mechanism, though effective for interpretability, introduces computational complexity that could be optimized with more efficient algorithms. Additionally, our comparative analysis between ISPD2015-B and ISPD2015-F indicates that performance on imbalanced datasets, while still superior to baselines, presents opportunities for improvement—a challenge shared across current congestion prediction approaches. Future work will address these aspects while maintaining MIHC's interpretability and prediction accuracy advantages.

**Broader impacts.** Our interpretable hypergraph model enhances EDA performance with limited impact. There is a minimal possibility that without proper validation, over-reliance risks erroneous decisions in chip design.

## Acknowledgements

The authors H.P., P.Z., X.X., and P.B. acknowledge the support by the U.S. Army Research Office (ARO) under Grant No. W911NF-23-1-0111, the National Science Foundation (NSF) under the NSF Award 2243104 under the Center for Complex Particle Systems (COMPASS), the Career Award CPS-1453860, Career Award 2337412, CNS-1932620, and NSF Mid-Career Advancement Award BCS-2527046, the Defense Advanced Research Projects Agency (DARPA) Young Faculty Award and DARPA Director Fellowship Award under Grant Number N66001-17-1-4044, Intel faculty awards and a Northrop Grumman grant. P.B. is also grateful to National Institute of Health (NIH) for the grants R01 AG 079957 "Interpretable machine learning to synergize brain age estimation and neuroimaging genetics" and RF1 AG 082201 "Neurovascular calcification and ADRD in two nonindustrial Native American populations". It was a wonderful experience designing and writing the grant application entitled "Neurovascular calcification and ADRD in two nonindustrial Native American populations" and awarded under RF1 AG 082201. The authors Z.Z. and X.L. acknowledge the support by NSFC (No. 72171229), the MOE Project of Key Research Institute of Humanities and Social Sciences (No. 22JJD110001), and Big Data and Responsible Artificial Intelligence for National Governance, RUC. The views, opinions, and/or findings in this article are those of the authors and should not be interpreted as official views or policies of the Department of Defense, the National Institute of Health or the National Science Foundation.

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

# A  Muti-View Information Bottleneck Loss Computation

As shown in Eq (4), we can wirte the objective of the cell-based hypergraphs $H^C$ to a loss function as follows:

$$\mathcal{L}^C = -\frac{1}{\beta^C} I(H^G; H_S^C) + I(H^C; H_S^C \mid H^G), \tag{16}$$

where $\beta^C$ is the trade-off parameter for $\mathcal{L}^C$, and we aim to minimize Eq (16) during the training. Similarly, the objective of the grid-based hypergraphs $H^G$ can be wirtten to a loss function as follows:

$$\mathcal{L}^G = -\frac{1}{\beta^G} I(H^C; H_S^G) + I(H^G; H_S^G \mid H^C), \tag{17}$$

where $\beta^G$ is the trade-off parameter for $\mathcal{L}^G$.

Then, we compute the average loss of $\mathcal{L}^C$ and $\mathcal{L}^G$, which helps us to optimize both $H_S^C$ and $H_S^G$:

$$L_{combination} = \frac{1}{2}\left(-\frac{1}{\beta^C} I(H^G; H_S^C) + I(H^C; H_S^C \mid H^G)\right) + \frac{1}{2}\left(-\frac{1}{\beta^G} I(H^C; H_S^G) + I(H^G; H_S^G \mid H^C)\right)$$

$$= \frac{1}{2}\left(I(H^C; H_S^C \mid H^G) + I(H^G; H_S^G \mid H^C)\right) - \frac{1}{2}\left(\frac{1}{\beta^C} I(H^G; H_S^C) + \frac{1}{\beta^G} I(H^C; H_S^G)\right).$$

For term $I(H^C; H_S^C \mid H^G)$, we can derive his upper bound. The conditional mutual information $I_\theta(H^C; H_S^C \mid H^G)$ is defined as:

$$I_\theta(H^C; H_S^C \mid H^G) = \mathbb{E}_{p(H^C, H^G)}\left[D_{\text{KL}}\left(p_\theta(H_S^C \mid H^C) \,\|\, p_\theta(H_S^C \mid H^G)\right)\right], \tag{18}$$

where the Kullback-Leibler (KL) divergence is given by:

$$D_{\text{KL}}(p\|q) = \mathbb{E}_{p(x)}\left[\log \frac{p(x)}{q(x)}\right]. \tag{19}$$

Substituting the definition of KL divergence into the conditional mutual information:

$$I_\theta(H^C; H_S^C \mid H^G) = \mathbb{E}_{H^C, H^G \sim p(H^C, H^G)}\mathbb{E}_{H_S \sim p_\theta(H_S^C \mid H^C)}\left[\log \frac{p_\theta(H_S^C \mid H^C)}{p_\theta(H_S^C \mid H^G)}\right]$$

$$= \mathbb{E}_{\mathbf{H}^C, \mathbf{H}^G \sim p(H^C, H^G)}\mathbb{E}_{\mathbf{H}_S \sim p_\theta(H_S^C \mid H^C)}\left[\log \frac{p_\theta(H_S^C = \mathbf{H}_S \mid H^C = \mathbf{H}^C)}{p_\theta(H_S^C = \mathbf{H}_S \mid H^G = \mathbf{H}^G)}\right].$$

Then, we derive the upper bound of $I_\theta\left(H^C; H_S^C \mid H^G\right)$:

$$I_\theta\left(H^C; H_S^C \mid H^G\right) = \mathbb{E}_{\mathbf{H}^C, \mathbf{H}^G \sim p(H^C, H^G)}\mathbb{E}_{\mathbf{H}_S \sim p_\theta(H_S^C \mid H^C)}\left[\log \frac{p_\theta(H_S^C = \mathbf{H}_S \mid H^C = \mathbf{H}^C)}{p_\theta(H_S^C = \mathbf{H}_S \mid H^G = \mathbf{H}^G)}\right]$$

$$= \mathbb{E}_{\mathbf{H}^C, \mathbf{H}^G \sim p(H^C, H^G)}\mathbb{E}_{\mathbf{H}_S \sim p_\theta(H_S^C \mid H^C)}\left[\frac{p_\theta(H_S^C = \mathbf{H}_S \mid H^C = \mathbf{H}^C)}{p_\gamma(H_S^G = \mathbf{H}_S \mid H^G = \mathbf{H}^G)} \cdot \frac{p_\gamma(H_S^G = \mathbf{H}_S \mid H^G = \mathbf{H}^G)}{p_\theta(H_S^C = \mathbf{H}_S \mid H^G = \mathbf{H}^G)}\right]$$

$$= D_{KL}\left(p_\theta(H_S^C \mid H^C) \,\|\, p_\gamma(H_S^G \mid H^G)\right) - D_{KL}\left(p_\theta(H_S^C \mid H^G) \,\|\, p_\gamma(H_S^G \mid H^G)\right)$$

$$\leq D_{KL}\left(p_\theta(H_S^C \mid H^C) \,\|\, p_\gamma(H_S^G \mid H^G)\right).$$

Similarly, we can get the upper bound for $I(H^G; H_S^G \mid H^C))$:

$$I(H^G; H_S^G \mid H^C)) \leq D_{KL}\left(p_\theta(H_S^G \mid H^G) \,\|\, p_\gamma(H_S^C \mid H^C)\right).$$

Therefore, the upper bound for the first term is:

$$\frac{1}{2}(I(H^C; H_S^C \mid H^G) + I(H^G; H_S^G \mid H^C))$$

$$\leq \frac{1}{2}(D_{KL}\left(p_\theta(H_S^C \mid H^C) \,\|\, p_\gamma(H_S^G \mid H^G)\right) + D_{KL}\left(p_\theta(H_S^G \mid H^G) \,\|\, p_\gamma(H_S^C \mid H^C)\right))$$

$$= D_{SKL}\left(p_\theta(H_S^C \mid H^C) \,\|\, p_\gamma(H_S^G \mid H^G)\right),$$

where $D_{SKL}(\cdot)$ denotes the symmetrized Kullback–Leibler (SKL) divergence.

Then, look at the $I(H^G; H_S^C)$ in the second term. According to the symmetry of mutual information, i.e., $I(X;Y) = I(Y;X)$, and the chain rule of mutual information, i.e., $I(X;Y,Z) = I(X;Y) + I(X;Z|Y)$, we can have:

$$I(H^G; H_S^C) = I(H_S^C; H^G) = I(H_S^C; H_S^G, H^G)$$
$$= I(H_S^C; H_S^G) + I(H_S^C; H^G | H_S^G).$$

Since the conditional mutual information is always non-negative, we can infer:

$$I(H_S^C; H^G | H_S^G) \geq 0. \tag{20}$$

Therefore, we can get the lower bound of $I(H^G; H_S^C)$ as:

$$I(H^G; H_S^C) \geq I(H_S^C; H_S^G). \tag{21}$$

Similarly, for $I(H^C; H_S^G)$, we can get,

$$I(H^C; H_S^G) \geq I(H_S^C; H_S^G). \tag{22}$$

Therefore, the second term gets the upper bound:

$$-\frac{1}{2}\left(\frac{1}{\beta^C} I(H^G; H_S^C) + \frac{1}{\beta^G} I(H^C; H_S^G)\right)$$
$$\leq -\frac{1}{2}\left(\frac{1}{\beta^C} I(H_S^C; H_S^G) + \frac{1}{\beta^G} I(H_S^C; H_S^G)\right)$$
$$= -\frac{\beta^C + \beta^G}{2\beta^C \beta^G} I(H_S^C; H_S^G).$$

Therefore, the combination loss has an upper bound:

$$L_{combination} \leq D_{SKL}\left(p_\theta(H_S^C \mid H^C) \parallel p_\gamma(H_S^G \mid H^G)\right) - \frac{\beta^C + \beta^G}{2\beta^C \beta^G} I(H_S^C; H_S^G).$$

By scaling both terms with $\beta = \frac{2\beta^C \beta^G}{\beta^C + \beta^G}$ and re-parameterizing the objective, we derive a tractable loss function for the MVIB framework:

$$\mathcal{L}_{MVIB} = -I(H_S^C; H_S^G) + \beta D_{SKL}\left(p_\theta(H_S^C \mid H^C) \parallel p_\gamma(H_S^G \mid H^G)\right), \tag{23}$$

where $p_\theta(H_S^C \mid H^C)$ and $p_\gamma(H_S^G \mid H^G)$ represent the bottleneck subgraph extractors for $H^C$ and $H^G$ parameterized by $\theta$ and $\gamma$ respectively, and $D_{SKL}(\cdot)$ denotes the symmetrized Kullback–Leibler (SKL) divergence. The hyperparameter $\beta$ controls the trade-off between the two terms.

## B  Mathematical Notations

Tables 5 and 6 summarize the mathematical notations used throughout the MIHC framework. Table 5 covers the fundamental problem formulation and basic Information Bottleneck concepts, while Table 6 details the components specific to multi-view representation learning and bottleneck mechanisms. Together, these notations describe the dual hypergraph structures that enable balanced fusion of topological and geometric circuit information for interpretable congestion prediction.

## C  Experimental Details

### C.1  Code Availability.

Our code is available in this link: `https://github.com/hping666/MIHC`.

Table 5: Mathematical Notations I: Problem Formulation and Information Bottleneck Framework

| Notation | Description |
| --- | --- |
| $L$ | Number of message passing layers |
| $P$ | Grid-based layout representation, $\in \mathbb{R}^{M \times N \times K}$ |
| $M, N$ | Grid dimensions |
| $K$ | Number of features per grid cell |
| $G$ | Cell-based hypergraph representation, $(V, E)$ |
| $V$ | Set of nodes (cells) |
| $E$ | Set of hyperedges (nets) |
| $C^G$ | Grid congestion map, $\in \mathbb{R}^{M \times N}$ |
| $C^C$ | Cell congestion graph, $\in \mathbb{R}^{|V|}$ |
| $\hat{C}^G$ | Predicted grid congestion map |
| $\hat{C}^C$ | Predicted cell congestion graph |
| $f_\theta$ | Predictive model parameterized by $\theta$ |
| $\mathcal{L}_{\text{sup}}$ | Supervised loss function |
| $X$ | Input variable in Information Bottleneck framework |
| $Y$ | Target variable in Information Bottleneck framework |
| $T$ | Intermediate representation in Information Bottleneck framework |
| $I(T; X)$ | Mutual information between $T$ and $X$ |
| $I(T; Y)$ | Mutual information between $T$ and $Y$ |
| $\beta^*$ | Trade-off parameter in Information Bottleneck framework |
| $H^C$ | Cell-based hypergraph |
| $H^G$ | Grid-based hypergraph |
| $V^C$ | Node set in cell-based hypergraph |
| $E^C$ | Hyperedge set in cell-based hypergraph |
| $X_v^C$ | Feature of node $v$ in cell-based hypergraph |
| $V^G$ | Node set in grid-based hypergraph |
| $E^G$ | Hyperedge set in grid-based hypergraph |
| $X_v^G$ | Feature of node $v$ in grid-based hypergraph |

## C.2 Hypergraph Construction Details

As shown in Figure 1(a) of our paper, we construct hypergraphs from circuit data through a specific process. In the layout example of Figure 1(a), we have 10 circuit components (cells), each represented as a node in the cell-based hypergraph. The layout is simultaneously divided into 9 grid cells, with each grid cell becoming a node in the grid-based hypergraph. The circuit nets connecting multiple circuit components are modeled as hyperedges in both representations. For the cell-based hypergraph, cells connected by the same net form a hyperedge. As illustrated in Figure 1(c) in the top-left diagram, blue cells 2, 5, 6, and 7 are connected by the same net, thus forming one hyperedge in the cell-based hypergraph.

For the grid-based hypergraph, grid cells traversed by the same net are connected via a hyperedge. Additionally, spatially adjacent grid cells are also connected through hyperedges to preserve geometric proximity information. As shown in Figure 1(c) in the bottom-left diagram, green grid cells 1, 4, 5, and 6 are traversed by the same net, thus forming one hyperedge in the grid-based hypergraph. Simultaneously, adjacent grid cells like 1-2, 2-3, 4-5, etc., would form their own hyperedges based on spatial adjacency.

## C.3 ISPD-2015 Benchmark Dataset

The ISPD-2015 benchmark consists of 20 industrial designs from various applications, with circuit sizes ranging from 29K to 1.3M cells, as shown in Table 7. These designs include both standard cells and macro blocks, presenting realistic challenges in placement and routing congestion prediction. Each design in the dataset includes:

- Circuit netlist information with both standard cells and macro blocks
- Placement solutions generated using Cadence Innovus v17.1 with varying parameters

Table 6: Mathematical Notations II: Multi-View Representation Learning and Bottleneck Components

| Notation | Description |
|---|---|
| $H_S^C$ | Bottleneck subgraph of cell-based hypergraph |
| $H_S^G$ | Bottleneck subgraph of grid-based hypergraph |
| $\beta^C$ | Trade-off parameter for cell-based Information Bottleneck |
| $p_\theta, p_\gamma$ | Probability model for bottleneck subgraph extraction |
| $\mathbf{M}^C$ | Incidence matrix of cell-based hypergraph |
| $\mathbf{X}^C$ | Node features of cell-based hypergraph |
| $\mathbf{p}$ | Probability vector for node selection |
| $\mathbf{X}_S^C$ | Node features of cell-based bottleneck subgraph |
| $\mathcal{N}(v^G)$ | Cell nodes associated with grid node $v^G$ |
| $W^C, W^G$ | Learnable weights for feature reweighting |
| $\mathcal{L}_{MVIB}$ | Multi-View Information Bottleneck loss |
| $D_{SKL}$ | Symmetrized KL divergence |
| $h_e^C$ | Hyperedge embedding in cell-based hypergraph |
| $h_e^G$ | Hyperedge embedding in grid-based hypergraph |
| $\mathcal{V}^C(e)$ | Nodes in cell-based hyperedge $e$ |
| $\mathcal{V}^G(e)$ | Nodes in grid-based hyperedge $e$ |
| $h_e^N$ | Net-based hyperedge embedding |
| $\hat{h}_v^C$ | Intermediate cell-based node embedding |
| $\hat{h}_v^N$ | Intermediate net-based node embedding |
| $\mathcal{E}^C(v)$ | Cell-based hyperedges containing node $v$ |
| $\mathcal{E}^N(v)$ | Net-based hyperedges containing node $v$ |
| $h_v^{C'}$ | Updated cell-based node embedding |
| $\hat{h}_v^G$ | Intermediate grid-based node embedding |
| $\mathcal{N}^G(v)$ | Spatially adjacent grid nodes to node $v$ |
| $h_v^{G'}$ | Updated grid-based node embedding |
| $h_S^C$ | Cell-based subgraph embedding |
| $h_S^G$ | Grid-based subgraph embedding |
| $\mathcal{L}_{\text{cont}}$ | Contrastive loss |
| $\tau$ | Temperature parameter in contrastive loss |
| $\hat{y}_v^C$ | Predicted cell-based node congestion |
| $\hat{y}_v^G$ | Predicted grid-based node congestion |
| $\mathcal{L}_{\text{total}}$ | Total loss function |

- Ground truth congestion maps derived from global routing solutions

The dataset provides several key features for congestion prediction:

- RUDY (Rectangular Uniform wire DensitY): Models the routing demand within bounding boxes of nets
- PinRUDY: Represents pin density distribution in grid cells
- MacroRegion: Binary indicators for regions covered by macro cells
- Horizontal/Vertical MacroMargin: Measures distances between margins of adjacent macros

For experimentation purposes, 533 different placement solutions were generated for each design using Innovus with varying parameters, resulting in a comprehensive dataset for training and evaluating congestion prediction models. The dataset divides the circuit layout into an M × N grid, with each grid cell represented as a pixel containing multiple feature channels. The routing overflow ground truth is provided in both horizontal and vertical directions.

### C.4 CircuitNet-N28 Benchmark Dataset

The CircuitNet-N28 is a comprehensive open-source dataset specifically designed for machine learning applications in VLSI CAD, particularly for cross-stage prediction tasks. It contains 10242

Table 7: Detailed Information of ISPD-2015 Benchmark Designs

| Design | #Cells | #Nets | Category |
|---|---|---|---|
| des_perf_1 | 113K | 113K | Medium |
| des_perf_a | 109K | 110K | Medium |
| des_perf_b | 113K | 113K | Medium |
| edit_dist_a | 130K | 131K | Medium |
| fft_1 | 35K | 33K | Small |
| fft_2 | 35K | 33K | Small |
| fft_a | 34K | 32K | Small |
| fft_b | 34K | 32K | Small |
| matrix_mult_1 | 160K | 159K | Large |
| matrix_mult_2 | 160K | 159K | Large |
| matrix_mult_a | 154K | 154K | Large |
| matrix_mult_b | 151K | 152K | Large |
| matrix_mult_c | 151K | 152K | Large |
| pci_bridge32_a | 30K | 30K | Small |
| pci_bridge32_b | 29K | 29K | Small |
| superblue11_a | 954K | 936K | Very Large |
| superblue12 | 1.3M | 1.3M | Very Large |
| superblue14 | 634K | 620K | Very Large |
| superblue16_a | 698K | 697K | Very Large |
| superblue19 | 522K | 512K | Very Large |

layout samples extracted from versatile runs of commercial design tools based on six open-source RISC-V designs in a 28nm technology node, as shown in Table 8.

- Circuit data is generated from six RISC-V designs (RISCY, RISCY-FPU, zero-riscy, and their variants)
- Layouts are generated using commercial tools (Cadence Innovus v20.10) with diverse parameter settings
- Ground truth labels include congestion, DRC violations, and IR drop information

The dataset provides several key features for prediction tasks:

- Macro Region: Regions covered by macros which are fixed during placement
- Cell Density: Cell distribution in each tile
- RUDY & Pin RUDY: Routing demand estimation for nets and pin density
- Pin Configuration: High-resolution representation of pin and routing blockage patterns
- Congestion: Overflow of routing demand in each tile

CircuitNet-N28 introduces significant diversity through various design parameters, including five core utilization settings (70%-90%), three macro placement strategies, eight power mesh combinations, and different filler insertion approaches. This results in 240 combinations of settings for each netlist, creating a dataset with substantial variation that better reflects real-world design scenarios. The tile size for all feature maps is 1.5 $\mu$m × 1.5 $\mu$m, aligning with the global routing cell size in commercial tools.

Table 8: Detailed Information of CircuitNet-N28 Benchmark Designs

| Design | #Cells | #Nets | Cell Area ($\mu m^2$) |
|---|---|---|---|
| RISCY-a | 45,717 | 47,759 | 65,739 |
| RISCY-FPU-a | 65,793 | 68,351 | 75,985 |
| zero-riscy-a | 34,299 | 43,970 | 58,631 |
| RISCY-b | 31,311 | 33,970 | 69,979 |
| RISCY-FPU-b | 51,126 | 54,327 | 80,030 |
| zero-riscy-b | 20,946 | 22,692 | 62,648 |

## C.5 Data Split Strategies

We employ two distinct data partitioning strategies to comprehensively evaluate our model's performance and generalization capability: placement-level split and design-level split.

### C.5.1 Placement-level Split

The placement-level split follows a standard design-level partitioning strategy where all designs appear in both training and testing sets, but with different placement solutions. As shown in Table 7, the ISPD2015 dataset contains 20 circuit designs, each with multiple placement solutions generated using different synthesis and placement parameters. For each design, we divide its corresponding placements into training and testing sets with a 7:3 ratio. This results in:

- **ISPD2015-F**: 374 training samples and 159 testing samples (all 20 designs including superblue series, 533 total placements)
- **ISPD2015-B**: 269 training samples and 114 testing samples (15 designs excluding superblue series, 383 total placements)

For CircuitNet-N28 (Table 8), we follow a similar approach across its 10,242 samples from six RISC-V designs, resulting in 7,170 training samples and 3,072 testing samples, maintaining the 7:3 ratio. This approach ensures all designs appear in both training and testing sets, allowing for comprehensive evaluation of model performance on known design types.

### C.5.2 Design-level Split for Generalization Evaluation

To rigorously evaluate our model's generalization capability to entirely unseen circuit designs, we conduct additional experiments following the design-level split setting. Unlike the placement-level split, this setting ensures training and test sets contain completely different circuit designs with no overlap between design families, providing a more stringent evaluation that better reflects real-world scenarios where models must generalize to new designs during deployment.

**ISPD2015-B Design Partitioning**    For ISPD2015-B containing 15 designs, we partition them into two groups ensuring designs with the same prefix remain together:

- **Group A** (8 designs): des_perf_1, des_perf_a, des_perf_b, fft_1, fft_2, fft_a, fft_b, edit_dist_a
- **Group B** (7 designs): matrix_mult_1, matrix_mult_2, matrix_mult_a, matrix_mult_b, matrix_mult_c, pci_bridge32_a, pci_bridge32_b

**ISPD2015-F Design Partitioning**    For ISPD2015-F containing 20 designs, we use a 10 vs 10 split:

- **Group A** (10 designs): des_perf series (3 designs: des_perf_1, des_perf_a, des_perf_b), fft series (4 designs: fft_1, fft_2, fft_a, fft_b), edit_dist_a, superblue12, superblue16_a
- **Group B** (10 designs): matrix_mult series (5 designs: matrix_mult_1, matrix_mult_2, matrix_mult_a, matrix_mult_b, matrix_mult_c), pci_bridge32 series (2 designs: pci_bridge32_a, pci_bridge32_b), superblue11_a, superblue14, superblue19

This partitioning strategy ensures that:

1. Designs with the same prefix (indicating functional similarity) remain in the same group, preventing information leakage between functionally related circuits
2. Both groups contain a mix of different design families and sizes
3. The evaluation captures the model's true capability to handle entirely unseen circuit architectures

**Bidirectional Evaluation**    We conduct bidirectional evaluation to provide comprehensive assessment:

1. **A→B**: Train on all placement solutions from Group A designs, test on all placement solutions from Group B designs

2. **B→A**: Train on all placement solutions from Group B designs, test on all placement solutions from Group A designs

This bidirectional approach ensures our evaluation is not biased by the specific choice of training/testing groups and provides robust evidence of generalization performance across different design families.

## C.6   Evaluation Metrics Details

This study employs a variety of evaluation metrics to assess the performance of the proposed model, including NMAE, NRMS, Pearson correlation, Spearman correlation, Kendall's tau for chip congestion prediction and Accuracy (ACC) for graph anomaly detection. Below are the definitions and mathematical formulations of each metric:

**Normalized Mean Absolute Error (NMAE):** NMAE measures the average absolute difference between predicted and true values, normalized by the range of the true values. It is defined as:

$$\text{NMAE} = \frac{1}{n} \sum_{i=1}^{n} \frac{|y_i - \hat{y}_i|}{\max(y) - \min(y)},$$

where $y_i$ is the true value, $\hat{y}_i$ is the predicted value, and $\max(y)$ and $\min(y)$ represent the maximum and minimum values of the true data, respectively. A lower NMAE indicates better predictive accuracy.

**Normalized Root Mean Square Error (NRMS):** NRMS quantifies the deviation between the predicted and actual values, normalized by the range of the true values. It is computed as:

$$\text{NRMS} = \frac{1}{\max(y) - \min(y)} \sqrt{\frac{1}{n} \sum_{i=1}^{n} (y_i - \hat{y}_i)^2}.$$

This metric penalizes larger errors more heavily, making it useful for evaluating prediction accuracy.

**Pearson Correlation Coefficient (Pearson):** The Pearson correlation coefficient measures the linear relationship between the predicted and true values, with values ranging from -1 (perfect negative correlation) to 1 (perfect positive correlation). It is given by:

$$\text{Pearson} = \frac{\sum_{i=1}^{n} (y_i - \bar{y})(\hat{y}_i - \bar{\hat{y}})}{\sqrt{\sum_{i=1}^{n} (y_i - \bar{y})^2 \sum_{i=1}^{n} (\hat{y}_i - \bar{\hat{y}})^2}},$$

where $\bar{y}$ and $\bar{\hat{y}}$ represent the mean of the true and predicted values, respectively.

**Spearman Rank Correlation (Spearman):** Spearman's rank correlation coefficient assesses the monotonic relationship between the true and predicted values by considering their ranks. It is calculated as:

$$\text{Spearman} = 1 - \frac{6 \sum_{i=1}^{n} d_i^2}{n(n^2 - 1)},$$

where $d_i$ is the difference in ranks for each pair of corresponding values, and $n$ is the number of data points.

**Kendall's Tau (Kendall):** Kendall's tau is another measure of rank correlation, specifically focusing on the concordance and discordance between pairs of predictions. It is defined as:

$$\text{Kendall} = \frac{C - D}{\frac{n(n-1)}{2}},$$

where $C$ is the number of concordant pairs, $D$ is the number of discordant pairs, and $n$ is the number of data points.

## C.7   Parameter Setting

In this section, we present the key implementation details of our model. The detailed parameter settings are listed in Table 9.

The key parameters include:

Table 9: Model Parameter Settings

| Parameter | Value | Parameter | Value |
|---|---|---|---|
| Hidden Dimension | 128 | Learning Rate | 1e-4 |
| # MV-HGNN Layers | 4 | Batch Size | 4 |
| # MLP Layers | 2 | Temperature $\tau$ | 0.07 |
| IB Weight $\beta$ | 0.1 | Dropout Rate | 0.1 |
| # Attention Heads | 4 | Optimizer | Adam |

- **Hidden Dimension**: The dimension of node embeddings in both cell-based and grid-based views.

- **# MV-HGNN Layers**: Number of Multi-View Heterogeneous Graph Neural Network layers.

- **# MLP Layers**: Number of layers in the prediction module MLPs.

- **IB Weight** $\beta$: The trade-off parameter in Information Bottleneck loss.

- **Temperature** $\tau$: Temperature parameter in contrastive learning.

- **# Attention Heads**: Number of attention heads in message passing.

The model is trained using the Adam optimizer with a learning rate of 1e-4. We apply dropout with rate 0.1 to prevent overfitting. The batch size is set to 32 for training efficiency.

## C.8 Time Consumption and Memory Usage

### C.8.1 Time Consumption

Our experiments on one NVIDIA A100 80GB GPU show practical performance across circuit scales. Small circuits ( 30K cells/nets) require only  0.3 seconds for inference and  0.9 seconds per training iteration, while very large circuits (1.3M cells/nets) take  1.6 seconds for inference and  5.0 seconds per training iteration. These measurements confirm that despite the model's sophistication, its computational demands remain manageable for real-world industrial applications, with reasonable inference times even for the largest circuits.

### C.8.2 Memory Usage

Our model's memory footprint scales efficiently with circuit size. Small circuits ( 30K cells/nets) consume approximately  3GB of GPU memory, while even the largest circuits (1.3M cells/nets) require  22GB during training, well within the 80GB capacity of modern A100 GPUs, ensuring our approach remains practical for industrial-scale designs.

# D   Result Supplement

## D.1   Chip Congestion Prediction Results

Table 10: Cell based Results on ISPD2015-B. The best in **bold** and the second underlined.

| Model | ISPD2015-B | | | | |
|---|---|---|---|---|---|
| | NMAE↓ | NRMS↓ | Pearson↑ | Spearman↑ | Kendall↑ |
| GCN | 0.038 | 0.044 | 0.547 | 0.498 | 0.421 |
| GAT | 0.039 (-2.63%) | 0.045 (-2.27%) | 0.531 (-2.94%) | 0.517 (+3.81%) | 0.463 (+9.95%) |
| HGNN | 0.036 (+5.26%) | 0.042 (+4.55%) | 0.577 (+5.50%) | 0.553 (+11.05%) | 0.476 (+13.07%) |
| CircuitGNN | 0.034 (+10.53%) | 0.040 (+9.09%) | 0.598 (+9.33%) | 0.611 (+22.68%) | 0.487 (+15.68%) |
| LHNN | 0.033 (+13.16%) | 0.038 (+13.64%) | 0.629 (+14.99%) | 0.627 (+25.90%) | 0.502 (+19.24%) |
| DE-HNN | 0.032 (+15.79%) | 0.037 (+15.91%) | 0.645 (+18.00%) | 0.632 (+26.91%) | 0.520 (+23.56%) |
| **MIHC(Ours)** | **0.029 (+23.68%)** | **0.034 (+22.73%)** | **0.687 (+25.66%)** | **0.689 (+38.43%)** | **0.574 (+36.38%)** |

Table 11: Grid based Results on ISPD2015-B and ISPD2015-F. The best in **bold** and the second underlined.

| Model | ISPD2015-B | | | | |
| --- | --- | --- | --- | --- | --- |
| | NMAE↓ | NRMS↓ | Pearson↑ | Spearman↑ | Kendall↑ |
| GCN | 0.043 | 0.049 | 0.405 | 0.162 | 0.137 |
| GAT | 0.044 (-2.33%) | 0.050 (-2.04%) | 0.398 (-1.73%) | 0.144 (-11.12%) | 0.141 (+2.92%) |
| HGNN | 0.040 (+6.98%) | 0.046 (+6.12%) | 0.451 (+11.36%) | 0.184 (+13.58%) | 0.142 (+3.65%) |
| RouteNet | 0.038 (+11.63%) | 0.045 (+8.16%) | 0.522 (+28.89%) | 0.198 (+22.22%) | 0.157 (+14.60%) |
| CircuitGNN | 0.037 (+13.95%) | 0.043 (+12.24%) | 0.561 (+38.39%) | 0.201 (+24.07%) | 0.164 (+19.71%) |
| CircuitFormer | 0.034 (+20.93%) | 0.040 (+18.37%) | 0.649 (+60.74%) | 0.231 (+42.59%) | 0.169 (+23.36%) |
| LHNN | 0.033 (+23.26%) | 0.040 (+18.37%) | 0.651 (+60.74%) | 0.252 (+55.56%) | 0.217 (+58.39%) |
| Lay-Net | 0.031 (+27.91%) | 0.037 (+24.49%) | 0.667 (+64.57%) | **0.301 (+85.19%)** | 0.248 (+81.76%) |
| **MIHC (Ours)** | **0.030 (+30.23%)** | **0.036 (+26.53%)** | **0.675 (+66.67%)** | 0.297 (+83.95%) | **0.252 (+83.94%)** |

Table 12: Cell based Results on ISPD2015-F. The best in **bold** and the second underlined.

| Model | ISPD2015-F | | | | |
| --- | --- | --- | --- | --- | --- |
| | NMAE↓ | NRMS↓ | Pearson↑ | Spearman↑ | Kendall↑ |
| GCN | 0.051 | 0.058 | 0.344 | 0.336 | 0.309 |
| GAT | 0.050 (+1.96%) | 0.057 (+1.72%) | 0.351 (+2.03%) | 0.342 (+1.79%) | 0.298 (-3.55%) |
| HGNN | 0.045 (+11.76%) | 0.052 (+10.34%) | 0.410 (+19.21%) | 0.366 (+8.93%) | 0.309 (0.00%) |
| CircuitGNN | 0.046 (+9.80%) | 0.053 (+8.62%) | 0.402 (+16.85%) | 0.378 (+12.50%) | 0.316 (+2.27%) |
| LHNN | 0.043 (+15.69%) | 0.049 (+15.52%) | 0.448 (+30.23%) | 0.384 (+14.29%) | 0.364 (+17.80%) |
| DE-HNN | 0.042 (+17.65%) | 0.048 (+17.24%) | 0.467 (+35.80%) | 0.395 (+17.57%) | 0.381 (+23.32%) |
| **MIHC(Ours)** | **0.035 (+31.37%)** | **0.041 (+29.31%)** | **0.524 (+52.33%)** | **0.512 (+52.38%)** | **0.447 (+44.64%)** |

Table 13: Grid based Results on ISPD2015-F. The best in **bold** and the second underlined.

| Model | ISPD2015-F | | | | |
| --- | --- | --- | --- | --- | --- |
| | NMAE↓ | NRMS↓ | Pearson↑ | Spearman↑ | Kendall↑ |
| GCN | 0.046 | 0.053 | 0.357 | 0.152 | 0.131 |
| GAT | 0.048 (+4.35%) | 0.055 (+3.77%) | 0.331 (-7.28%) | 0.137 (-9.87%) | 0.116 (-11.47%) |
| HGNN | 0.043 (+6.52%) | 0.050 (+5.66%) | 0.352 (-1.40%) | 0.147 (-3.29%) | 0.121 (-7.63%) |
| RouteNet | 0.042 (+8.70%) | 0.050 (+5.66%) | 0.362 (+1.40%) | 0.161 (+5.92%) | 0.137 (+4.58%) |
| CircuitGNN | 0.042 (+8.70%) | 0.049 (+7.55%) | 0.364 (+2.00%) | 0.162 (+6.58%) | 0.134 (+2.29%) |
| CircuitFormer | 0.038 (+17.39%) | 0.045 (+15.09%) | 0.453 (+26.91%) | 0.196 (+28.95%) | 0.165 (+26.02%) |
| LHNN | 0.037 (+19.57%) | 0.044 (+16.98%) | 0.464 (+29.97%) | 0.221 (+45.39%) | 0.182 (+38.93%) |
| Lay-Net | 0.035 (+23.91%) | 0.042 (+20.75%) | 0.484 (+35.53%) | 0.256 (+68.42%) | 0.208 (+58.02%) |
| **MIHC (Ours)** | **0.032 (+30.43%)** | **0.039 (+26.42%)** | **0.503 (+41.00%)** | **0.271 (+78.95%)** | **0.227 (+73.25%)** |

Table 14: Cell based Results on CircuitNet-N28. The best in **bold** and the second underlined.

| Model | CircuitNet-N28 | | | | |
| --- | --- | --- | --- | --- | --- |
| | NMAE↓ | NRMS↓ | Pearson↑ | Spearman↑ | Kendall↑ |
| GCN | 0.045 | 0.051 | 0.516 | 0.438 | 0.319 |
| GAT | 0.046 (-2.22%) | 0.051 (+0.00%) | 0.513 (-0.58%) | 0.447 (+2.05%) | 0.321 (+0.63%) |
| HGNN | 0.044 (+2.22%) | 0.048 (+5.88%) | 0.552 (+6.98%) | 0.468 (+6.85%) | 0.332 (+4.08%) |
| CircuitGNN | 0.040 (+11.11%) | 0.043 (+15.69%) | 0.609 (+18.02%) | 0.500 (+14.16%) | 0.368 (+15.36%) |
| LHNN | 0.039 (+13.33%) | 0.043 (+15.69%) | 0.617 (+19.57%) | 0.509 (+16.21%) | 0.366 (+14.73%) |
| DE-HNN | 0.038 (+15.56%) | 0.042 (+17.65%) | 0.638 (+23.64%) | 0.511 (+16.67%) | 0.374 (+17.24%) |
| **MIHC(Ours)** | **0.036 (+20.00%)** | **0.039 (+23.53%)** | **0.661 (+28.10%)** | **0.522 (+19.18%)** | **0.396 (+24.14%)** |

## D.2 Expalination Results

Due to the lack of publicly available, interpretable datasets in the EDA domain, we use the commonly adopted MNIST 0/1 dataset to demonstrate the explainability of our proposed MIHC. To validate the effectiveness of the explainability module in our model, we conduct explainability testing experiments on these datasets to test our model's capability in identifying important nodes in the graph through node-level explainability testing. The results of these tests are presented in Table 16. Our proposed method demonstrates strong consistency across both datasets, exhibiting high performance across various evaluation metrics. This suggests that our approach provides reliable and coherent explanations across different types of data, indicating that the model's interpretability is robust and can generalize well. The consistent high scores in all the key metrics underscore the model's ability to

Table 15: Grid based Results on CircuitNet-N28. The best in **bold** and the second underlined.

| Model | CircuitNet-N28 | | | | |
| | NMAE↓ | NRMS↓ | Pearson↑ | Spearman↑ | Kendall↑ |
|---|---|---|---|---|---|
| GCN | 0.052 | 0.053 | 0.417 | 0.399 | 0.312 |
| GAT | 0.052 (+0.00%) | 0.054 (-1.89%) | 0.411 (-1.44%) | 0.405 (+1.50%) | 0.323 (+3.53%) |
| HGNN | 0.051 (+1.92%) | 0.052 (+1.89%) | 0.442 (+6.00%) | 0.424 (+6.27%) | 0.331 (+6.09%) |
| RouteNet | 0.047 (+9.62%) | 0.049 (+7.55%) | 0.541 (+29.74%) | 0.501 (+25.56%) | 0.373 (+19.55%) |
| CircuitGNN | 0.047 (+9.62%) | 0.048 (+9.43%) | 0.547 (+31.18%) | 0.508 (+27.32%) | 0.371 (+18.91%) |
| CircuitFormer | 0.044 (+15.38%) | 0.046 (+13.21%) | 0.622 (+49.16%) | 0.526 (+31.83%) | 0.384 (+23.08%) |
| LHNN | 0.043 (+17.31%) | 0.044 (+16.98%) | 0.646 (+54.92%) | 0.533 (+33.58%) | 0.389 (+24.68%) |
| Lay-Net | 0.041 (+21.15%) | 0.043 (+18.87%) | 0.671 (+60.91%) | 0.558 (+39.85%) | **0.406 (+30.13%)** |
| **MIHC (Ours)** | **0.040 (+23.08%)** | **0.041 (+22.64%)** | **0.682 (+63.55%)** | **0.572 (+43.36%)** | 0.402 (+28.85%) |

Table 16: Node explanation test on MNIST0 and MNIST1. The best in **bold** and the second underlined.

| Dataset | Method | ACC | Precision | Recall | F1 | AUC |
|---|---|---|---|---|---|---|
| MNIST0 | ours | **0.742** | 0.669 | 0.725 | **0.696** | **0.697** |
| | GCN+GNNE | 0.527 | 0.343 | 0.500 | 0.407 | 0.554 |
| | GAT+GNNE | 0.419 | 0.349 | **0.917** | 0.506 | 0.689 |
| | SIGNET | 0.650 | **0.995** | 0.041 | 0.078 | 0.615 |
| MNIST1 | ours | **0.721** | 0.642 | 0.698 | **0.669** | **0.685** |
| | GCN+GNNE | 0.527 | 0.359 | 0.583 | 0.444 | 0.573 |
| | GAT+GNNE | 0.419 | 0.321 | **0.708** | 0.442 | 0.594 |
| | SIGNET | 0.650 | **1.000** | 0.041 | 0.079 | 0.522 |

offer actionable and meaningful insights for decision-making in practical applications. In comparison, baseline methods show significant limitations. GAT+GNNE suffers from low precision despite high recall, while SIGNET achieves high precision but fails to capture comprehensive node relationships, leading to low recall scores. The core principle of explainability—identifying the most important part of the graph—remains the same across both MNIST and chip congestion scenarios.

Furthermore, to demonstrate MIHC's explainability under realistic congestion scenarios, we present a chip congestion heatmap in Figure 2. This visualization highlights the model's capability to accurately localize critical regions on the chip. As clearly shown, MIHC achieves the closest match to the ground truth, underscoring its superior interpretability.

### D.3 Hyper-parameter Sensitivity

Detailed results of hyper-parameter sensitivity are shown in Table 17 and Table 18.

Table 17: Hidden Dimension Sensitivity Analysis on ISPD2015-B

| Hidden Dim | Cell-based Results | | | | | Grid-based Results | | | | |
| | NMAE↓ | NRMS↓ | Pearson↑ | Spearman↑ | Kendall↑ | NMAE↓ | NRMS↓ | Pearson↑ | Spearman↑ | Kendall↑ |
|---|---|---|---|---|---|---|---|---|---|---|
| 32 | 0.036 | 0.040 | 0.579 | 0.592 | 0.484 | 0.041 | 0.045 | 0.477 | 0.199 | 0.151 |
| 64 | 0.030 | 0.036 | 0.663 | 0.668 | 0.551 | 0.032 | 0.037 | 0.662 | 0.281 | 0.246 |
| 128 | 0.029 | 0.034 | 0.687 | 0.689 | 0.574 | 0.030 | 0.036 | 0.675 | 0.297 | 0.252 |
| 256 | 0.029 | 0.034 | 0.682 | 0.688 | 0.577 | 0.030 | 0.036 | 0.676 | 0.294 | 0.250 |

Table 18: Multi-view HGNN Layer Number Sensitivity Analysis on ISPD2015-B

| #Layers | Cell-based Results | | | | | Grid-based Results | | | | |
|---|---|---|---|---|---|---|---|---|---|---|
| | NMAE↓ | NRMS↓ | Pearson↑ | Spearman↑ | Kendall↑ | NMAE↓ | NRMS↓ | Pearson↑ | Spearman↑ | Kendall↑ |
| 1 | 0.033 | 0.039 | 0.612 | 0.619 | 0.492 | 0.039 | 0.044 | 0.510 | 0.192 | 0.148 |
| 2 | 0.030 | 0.035 | 0.679 | 0.672 | 0.566 | 0.031 | 0.036 | 0.672 | 0.290 | 0.251 |
| 3 | 0.029 | 0.034 | 0.690 | 0.681 | 0.571 | 0.030 | 0.036 | 0.672 | 0.294 | 0.252 |
| 4 | 0.029 | 0.034 | 0.687 | 0.689 | 0.574 | 0.030 | 0.036 | 0.675 | 0.297 | 0.252 |
| 5 | 0.030 | 0.034 | 0.688 | 0.684 | 0.571 | 0.030 | 0.037 | 0.671 | 0.288 | 0.247 |
| 6 | 0.030 | 0.035 | 0.671 | 0.666 | 0.554 | 0.031 | 0.037 | 0.669 | 0.288 | 0.248 |

