# OpenReview forum: "MIHC: Multi-View Interpretable Hypergraph Neural Networks with Information Bottleneck for Chip Congestion Prediction"
_NeurIPS.cc/2025/Conference — NeurIPS 2025 poster_

### Official Review · Reviewer_ERG1 · 2025-06-27

**Clarity:** 4
**Significance:** 3
**Originality:** 4
**Rating:** 5
**Confidence:** 4

**Summary:**

This work proposes MIHC, a new way of integrating both cell-based and grid-based circuit features to enhance congestion prediction. It also considers interpretability using an information bottleneck mechanism. The authors provide comprehensive experiments on a wide range of test cases. The tests on both ISPD2015 and CircuitNet are convincing.

**Questions:**

- Is there any chance that your method could provide a method for chip congestion optimization? Like congestion-driven placement?

**Ethical Concerns:**

["NO or VERY MINOR ethics concerns only"]

**Final Justification:**

I believe this work does push the boundary of chip congestion prediction and present convincing results. The authors' rebuttal addresses my concern of runtime. The authors have not presented a detailed and applicable way of integrating the method into a congestion-driven optimization flow. But it does not eliminate the value of the work itself.

As a conclusion, I vote for acceptance of this work.

**Limitations:**

Yes

**Quality:**

4

**Strengths And Weaknesses:**

Strengths:
- This paper gives a comprehensive and clear comparing analysis of the existing literatures. Chip congestion prediction is a vital problem in EDA community and worth investigating.
- The interpretable subgraph module for feature fusion is intuitive. The idea of adopting nets to connect not only graph nodes, but also grid cells is an interesting and reasonable design.
- The experimental results provide a large amount of baselines and superior performance.
- I like the part C.8 since it provides detailed runtime and memory use and all those statistics. That helps readers understand the computation resource immediately and can quickly assess whether this technique is applicable to their own designs and own device.

Weaknesses:
- Could you please detail the runtime comparison with other baseline methods?

---

> ### Author Rebuttal · Authors · 2025-07-30
>
> We sincerely thank the reviewer for taking the time to review our paper and for recognizing our contributions, including the clear comparative analysis of related work, the significance of chip congestion prediction, the intuitive design of the interpretable subgraph module, the comprehensive and strong experimental results, and the practical runtime and memory analysis. In the following, we provide thoughtful and thorough responses to all of the reviewer’s questions.
>
> >W1: Could you please detail the runtime comparison with other baseline methods?
>
> **A:**
> Thank you for the helpful question regarding runtime comparison. To provide a clear and fair evaluation, we selected the best-performing baseline methods from cell-based (DE-HNN) and grid-based (Lay-Net) approaches for runtime comparison on the same NVIDIA A100 80GB GPU setup:
>
> | Method | Small Circuits (~30K cells/nets) | Very Large Circuits (1.3M cells/nets) |
> |--------|----------------------------------|---------------------------------------|
> | MIHC (Ours) | 0.3s | 1.6s |
> | DE-HNN | 0.3s | 1.5s |
> | Lay-Net | 0.9s | 3.8s |
>
> From a comprehensive performance-runtime perspective, our MIHC achieves the best trade-off: it maintains competitive inference speed comparable to DE-HNN while delivering significantly superior prediction accuracy (16.67% NMAE reduction on ISPD2015-F) and providing interpretability capabilities that neither baseline offers. Compared to Lay-Net, our method achieves both better runtime efficiency and higher accuracy, making it more suitable for practical industrial applications.
>
>
> >Q1: Is there any chance that your method could provide a method for chip congestion optimization? Like congestion-driven placement?
>
> **A:**
> Thank you for the insightful question. While our current work focuses on congestion prediction, we believe that the architecture of MIHC is well suited to support congestion-aware design optimization tasks. In particular, MIHC’s ability to identify congestion-critical regions and jointly reason over netlist and layout information makes it a promising candidate for both chip-level optimization and placement refinement.
>
> **Chip Congestion Optimization:** Our Information Bottleneck mechanism identifies critical congestion-correlated regions (as shown in Figure 2), revealing not just where congestion occurs but which local structures cause these anomalies. The bottleneck subgraphs identified by our method could guide chip optimization by highlighting problematic regions that need cell redistribution or routing resource allocation. The interpretable nature of our predictions provides actionable guidance for designers to make targeted optimizations rather than generic congestion mitigation, enabling more efficient use of routing resources and improved overall chip performance.
>
> **Congestion-Driven Placement:** For placement-specific applications, our method could be integrated into placement algorithms to evaluate congestion impact during cell positioning. By simultaneously processing both netlist topology and layout geometry, our method can provide insights into how placement decisions affect congestion from both connectivity and spatial perspectives. This multi-view analysis capability allows placement tools to consider both topological connectivity constraints and geometric layout effects simultaneously, leading to more informed placement decisions that proactively avoid congestion hotspots rather than reactively addressing them post-placement.

---

> > ### Comment · Reviewer_ERG1 · 2025-08-06
> > **Thank you for your response.**
> >
> > Your response does answer my questions. I will keep my score.

---

> > > ### Author Response · Authors · 2025-08-06
> > > **Appreciation for your review**
> > >
> > > Thank you very much for your positive feedback and for maintaining the score.
> > >
> > > We greatly appreciate your recognition of our work and the valuable suggestions you provided during the review process. Your constructive comments have been instrumental in helping us improve our paper, and we are grateful for the time and effort you invested in providing such thoughtful feedback.
> > >
> > > We look forward to incorporating your insights into the revised version of our manuscript.

---

> ### Comment · Area_Chair_uRFj · 2025-08-05
>
> Dear Reviewer ERG1,
>
> Please join the discussion to assess whether the authors’ response has adequately addressed your question. Thank you very much for your engagement!
>
> Best regards,
> AC

---

### Official Review · Reviewer_8ndm · 2025-07-04

**Clarity:** 3
**Significance:** 3
**Originality:** 3
**Rating:** 4
**Confidence:** 3

**Summary:**

This paper proposes a multi-view hypergraph neural network framework for chip congestion prediction. This framework processes both graph and image information in hypergraph representations to fuse multi-view circuit data information. This framework also implements a subgraph Information Bottleneck mechanism to identify critical congestion-correlated regions. Experiments show this framework achieves great improvement on ISPD2025 and CircuitNet-N28.

**Questions:**

1. In equation (9),  $h_e^C$ and $h_e^G$ are concatenated. Does this mean cell-based and grid-based hypergraphs share the same topological structure (Because equation (9) concatenates the embedding of the same edge from two hypergraphs)? If not, how do you construct cell-based and grid-based hypergraphs to achieve this?

**Ethical Concerns:**

["NO or VERY MINOR ethics concerns only"]

**Final Justification:**

After reading the authors' rebuttal, my overall impression of the paper has improved, but I am keeping my original rating.
﻿
**Resolved issues:**
﻿
* **Interpretability (RQ2)**: The rebuttal provided clearer explanations and visualizations in the chip congestion prediction task, plus complementary results on the MNIST 0/1 benchmark. This better demonstrates how the Information Bottleneck mechanism extracts meaningful subgraphs.
* **Equation (9) and topology correspondence**: The authors clarified that the cell-based and grid-based hypergraphs have different topologies but share hyperedges representing the same nets, making concatenation meaningful. The example was convincing.
﻿
**Unresolved issues:**
﻿
* The additional computational complexity introduced by the Information Bottleneck mechanism remains unaddressed.
* Lower performance on imbalanced datasets (ISPD2025-F) compared to balanced datasets (ISPD2025-B) was not addressed.
﻿
**Weighting:**
The resolved issues significantly improve the paper’s clarity and technical soundness. However, the unresolved limitations still affect the completeness of the evaluation and practical applicability. These factors balance out to maintain the rating.

**Limitations:**

1. The Information Bottleneck mechanism introduces computational complexity.
﻿
2. Performance on imbalanced datasets (ISPD2025-F) is worse than on balanced datasets (ISPD2025-B).

**Paper Formatting Concerns:**

There seems to be no problem.

**Quality:**

3

**Strengths And Weaknesses:**

**Strengths**
﻿
1. MIHC naturally and effectively fuses the cell-based and grid-based information.

2. Chip congestion prediction results (RQ1) show this framework achieves great improvement on ISPD2015-B and CircuitNet-N28.

**Weaknesses**
1. The illustration of MIHC’s interpretability performance(RQ2) is not clear.

---

> ### Author Rebuttal · Authors · 2025-07-30
>
> We sincerely appreciate the reviewer for the positive feedback, especially on the effectiveness of MIHC’s fusion strategy and its strong performance on ISPD2015-B and CircuitNet-N28. We now provide detailed, point-by-point responses to the reviewer’s comments as follows.
>
> >W1: The illustration of MIHC’s interpretability performance(RQ2) is not clear.
>
> **A1:** Thank you for your valuable feedback. We recognize that the interpretability evaluation of MIHC (RQ2) can benefit from a clearer explanation. To address this, we demonstrate MIHC's interpretability performance through two complementary tasks: chip congestion prediction and the MNIST 0/1 classification benchmark.
>
> In the chip congestion prediction scenario (Figure 2 in the main paper), MIHC is evaluated on its ability to detect meaningful structural patterns that contribute to congestion. As illustrated in the heatmaps, the bottleneck subgraphs extracted by our model effectively capture critical regions where congestion occurs. These regions align closely with the ground truth patterns, confirming that MIHC not only detects individual anomalies but also identifies the local substructures responsible for those anomalies. This is especially important in circuit design, where congestion often arises from interactions among components rather than from isolated nodes. Our information bottleneck mechanism highlights these structurally significant areas, offering insights beyond what node-level detection alone can provide.
>
> To further support the interpretability of MIHC, we also include an evaluation on the MNIST 0/1 dataset (Table 15 in the Appendix). This dataset is commonly used in interpretability studies because it contains semantically meaningful explainable labels. Since interpretability benchmarks are lacking in the chip congestion domain, we use MNIST as a well-established complement. The results show that MIHC achieves consistently high scores across a range of evaluation metrics, demonstrating robustness and generalizability. The performance indicates that our method provides coherent and faithful explanations across different data types. Compared to baseline models, MIHC exhibits a better balance between precision and recall. For example, GAT+GNNE achieves high recall but low precision, producing noisy explanations, while SIGNET prioritizes precision but overlooks broader node relationships, resulting in poor recall. MIHC, by contrast, maintains both high fidelity and meaningful structural coverage.
>
> Overall, the core interpretability mechanism of MIHC—extracting the most informative subgraph responsible for predictions—remains consistent across both chip and MNIST scenarios. This unified explanation framework ensures that the insights generated are not only accurate but also actionable, making the model's decisions more transparent and trustworthy for real-world applications.
>
> >Q1: In equation (9), $h_e^C$ and $h_e^G$ are concatenated. Does this mean cell-based and grid-based hypergraphs share the same topological structure (Because equation (9) concatenates the embedding of the same edge from two hypergraphs)? If not, how do you construct cell-based and grid-based hypergraphs to achieve this?
>
> **A:**
> Thank you for this important clarification question. The cell-based and grid-based hypergraphs do **not** share the same topological structure, as they operate on different node types (cells vs. grid regions). However, they share corresponding hyperedges that represent the same nets, which enables meaningful concatenation in Equation (9).
>
> **Different Topologies, Same Net Correspondence:** While the two hypergraphs have different topological structures due to different node representations, their hyperedges correspond to identical nets in the circuit. This net-based correspondence is the key to our multi-view information fusion approach—we use net hyperedges as bridges to enable information exchange between the two hypergraph representations.
>
> **Construction Example:** As illustrated in Figure 1(c), consider the purple hyperedge 1: in the cell-based hypergraph, it connects blue cells 2, 5, 6, and 7 that are connected by the same net; in the grid-based hypergraph, it connects green grid cells 1, 4, 5, and 6 that are traversed by the same net. Although the connected nodes differ, both hyperedges represent the same physical net, making their embeddings ($h_e^C$ and $h_e^G$) complementary views of the same connectivity information.
>
> This net-based bridging mechanism allows us to meaningfully concatenate embeddings from structurally different hypergraphs while preserving the semantic correspondence between views.

---

> ### Comment · Area_Chair_uRFj · 2025-08-05
>
> Dear Reviewer 8ndm,
>
> Please join the discussion to assess whether the authors’ response has adequately addressed your questions and concerns. Thank you very much for your engagement!
>
> Best regards,
> AC

---

> ### Author Response · Authors · 2025-08-07
> **Request for Your Valuable Feedback on Our Rebuttal**
>
> Dear Reviewer 8ndm,
>
> Thank you for your thorough review and positive feedback on our paper about MIHC for chip congestion prediction. We deeply appreciate your recognition of our framework's effectiveness in fusing cell-based and grid-based information and the strong performance results on ISPD2015-B and CircuitNet-N28.
>
> We have addressed your concerns regarding **MIHC's interpretability performance illustration**, **the topological structure relationship between cell-based and grid-based hypergraphs**, and **the concatenation mechanism in equation (9)** in our detailed rebuttal. We particularly provided comprehensive explanations of how our net-based correspondence enables meaningful multi-view fusion despite different topological structures, along with additional MNIST evaluation results to demonstrate the robustness of our interpretability mechanism across different domains.
>
> We would be extremely grateful if you could share your thoughts on our response and let us know if our clarification of the hypergraph construction methodology and interpretability evaluation have adequately addressed your questions, or if there are any further concerns we could clarify.
>
> Thank you again for your time and consideration.
>
> Best regards,
>
> All the Authors

---

### Official Review · Reviewer_ibfY · 2025-07-08

**Clarity:** 3
**Significance:** 3
**Originality:** 2
**Rating:** 4
**Confidence:** 3

**Summary:**

This paper proposes a dual hyper graph approach to congestion prediction for chip design. Congestion prediction is an important problem because during the chip design process, when wiring congestion occurs, it can require a lot of disruptive changes to the design. If it can be predicted, it can be avoided earlier in the design process and therefore save a lot of time and money while improving the quality of results. However, predicting congestion is difficult, and although many prior works have made progress on this problem, it remains challenging. This paper proposes improving the congestion prediction model by using both a netlist hypergraph as well as a layout hypergraph in order to simultaneously understand the circuit as well as the proposed layout, combining them with an information bottleneck in order to better analyze the design. The proposed approach yields impressive results on standard academic benchmarks and also provides interpretability so that engineers can analyze the design and understand how congestion arises.

**Questions:**

Why is it ok to include subsets of the same designs in both training and test sets? "This approach ensures all designs appear in both training and testing sets, allowing for comprehensive evaluation of model generalization capability." - I'm concerned this reduces the ability to comprehensively evaluate model generation.

How often do you train a new model and how much compute does it take?

**Ethical Concerns:**

["NO or VERY MINOR ethics concerns only"]

**Final Justification:**

I raised my score to reflect the authors' rebuttal and extra experiments. I still don't feel the experiments are 100% convincing, which is why I'm giving a borderline accept score. But I do like the paper.

**Limitations:**

yes

**Quality:**

3

**Strengths And Weaknesses:**

The empirical results seem strong, and the high-level components seem reasonable. The paper is also clearly written.

The biggest weakness in my mind has to do with generalization. I wasn't clear whether the paper trains a single model and tries to generalize across all designs, or whether there were separate models trained for different designs. I had questions about the training/test split from the datasets the authors created, because generally it's easier for a model to make predictions about a design if very closely related circuits are in the training set. And in this case it appears the authors split up designs so that pieces of each design were both in the training set and the test set.
Additionally there are quite a few components to the model and it does seem a little complex. However, that is often the case in chip design methods.
Finally I'd like to see more about training cost and frequency.

---

> ### Author Rebuttal · Authors · 2025-07-30
>
> We thank the reviewer for carefully identifying the concern regarding generalization and training cost, which we address in detail below, point by point. We also appreciate the positive remarks on our empirical results, overall design, and writing clarity.
>
> >W1: The biggest weakness in my mind has to do with generalization. I wasn't clear whether the paper trains a single model and tries to generalize across all designs, or whether there were separate models trained for different designs.
>
> **A:** Thank you for bringing this concern about model generalization. To clarify, in this work, we train a single model for each dataset that generalizes across all designs within that dataset. Specifically, we obtain three models in total: one trained on ISPD2015-B, one on ISPD2015-F, and one on CircuitNet-N28. Each model is trained to handle multiple different circuit designs simultaneously. For example, ISPD2015-F contains 20 different industrial designs (as shown in Table 6), and our model learns to predict congestion across all these diverse designs with a single set of parameters.
>
> >W2 + Q1: I had questions about the training/test split from the datasets the authors created, because generally it's easier for a model to make predictions about a design if very closely related circuits are in the training set. And in this case it appears the authors split up designs so that pieces of each design were both in the training set and the test set.
>
> **A:**
> Thank you for raising this important and constructive question about our experimental setup. In our original experiments, we followed a placement-level split similar to Exp1 in Lay-Net [1], where different placement solutions from the same design appeared in both training and test sets. While this approach is useful for evaluating model performance on known design types, we notice that it could bring concerns about generalization capability since models might learn design-specific patterns rather than generalizable congestion prediction principles.
>
> To address your concern and provide a more rigorous evaluation of generalization capability, we have conducted additional experiments following the unseen design experimental setup (Exp2) from Lay-Net [1]. In this setting, we performed design-level splits where training and test sets contain completely different circuit designs, ensuring no overlap between design families. Due to the time limit, we focus on ISPD2015-B and ISPD2015-F datasets for these additional generalization experiments. For ISPD2015-B, we divided the 15 designs into Group A (8 designs: des_perf series, fft series, and edit_dist_a) and Group B (7 designs: matrix_mult series and pci_bridge32 series). For ISPD2015-F, we used a 10 vs 10 split, with Group A containing des_perf series, fft series, edit_dist_a, superblue12, and superblue16_a, while Group B included matrix_mult series, pci_bridge32 series, superblue11_a, superblue14, and superblue19. This partitioning ensures that designs with the same prefix remain together, preventing any potential issues between functionally similar circuits. Following the Lay-Net [1] Exp2 experimental setup, we conducted bidirectional evaluation by training on Group A and testing on Group B, then reversing the roles to train on Group B and test on Group A.
>
> The results from our unseen design experiments demonstrate significantly more challenging conditions, with substantial performance degradation across all methods compared to the seen design setting. As shown in the following experimental tables, which present the comprehensive results for both A→B and B→A training-testing directions across ISPD2015-B and ISPD2015-F datasets, MIHC maintains superior performance over state-of-the-art baselines in the vast majority of evaluation metrics across both cell-based and grid-based prediction tasks. For instance, in the B→A experiments, MIHC achieves the best performance in 18 out of 20 metrics, with only occasional slight disadvantages in specific cases such as Lay-Net's marginally better performance in grid-based Spearman correlation on ISPD2015-F. These results provide evidence that our method achieves genuine generalization capability rather than simply memorizing design-specific characteristics, offering a more realistic and stringent evaluation that better reflects real-world scenarios where models must generalize to entirely new circuit designs during practical deployment.
>
> **Table 1: Unseen Design Results - Group A Training → Group B Testing**
>
> | Model | ISPD2015-B |||||  ISPD2015-F |||||
> |-------|-------|-------|----------|-----------|----------|-------|-------|----------|-----------|----------|
> |       | NMAE↓ | NRMS↓ | Pearson↑ | Spearman↑ | Kendall↑ | NMAE↓ | NRMS↓ | Pearson↑ | Spearman↑ | Kendall↑ |
> | **Cell-based** |
> | CircuitGNN | 0.058 | 0.064 | 0.376 | 0.335 | 0.248 | 0.071 | 0.077 | 0.343 | 0.304 | 0.218 |
> | LHNN | 0.054 | 0.060 | 0.398 | 0.355 | 0.265 | 0.067 | 0.073 | 0.368 | 0.321 | 0.235 |
> | DE-HNN | 0.051 | 0.057 | 0.415 | 0.372 | 0.282 | 0.063 | 0.069 | 0.385 | 0.342 | 0.252 |
> | MIHC (Ours) | **0.048** | **0.053** | **0.448** | **0.405** | **0.302** | **0.059** | **0.065** | **0.414** | **0.375** | **0.277** |
> | **Grid-based** |
> | CircuitGNN | 0.061 | 0.067 | 0.366 | 0.158 | 0.132 | 0.074 | 0.080 | 0.331 | 0.148 | 0.122 |
> | LHNN | 0.057 | 0.063 | 0.389 | 0.172 | 0.145 | 0.070 | 0.076 | 0.349 | 0.162 | 0.135 |
> | Lay-Net | 0.053 | 0.059 | 0.411 | 0.192 | 0.165 | 0.064 | 0.070 | 0.394 | 0.200 | **0.174** |
> | MIHC (Ours) | **0.051** | **0.057** | **0.436** | **0.218** | **0.188** | **0.064** | **0.069** | **0.405** | **0.201** | 0.167 |
>
>
> **Table 2: Unseen Design Results - Group B Training → Group A Testing**
>
> | Model | ISPD2015-B |||||  ISPD2015-F |||||
> |-------|-------|-------|----------|-----------|----------|-------|-------|----------|-----------|----------|
> |       | NMAE↓ | NRMS↓ | Pearson↑ | Spearman↑ | Kendall↑ | NMAE↓ | NRMS↓ | Pearson↑ | Spearman↑ | Kendall↑ |
> | **Cell-based** |
> | CircuitGNN | 0.063 | 0.068 | 0.351 | 0.325 | 0.232 | 0.075 | 0.081 | 0.337 | 0.293 | 0.205 |
> | LHNN | 0.058 | 0.065 | 0.379 | 0.347 | 0.246 | 0.070 | 0.076 | 0.354 | 0.315 | 0.222 |
> | DE-HNN | 0.053 | 0.058 | 0.405 | 0.365 | 0.269 | 0.065 | 0.071 | 0.378 | 0.336 | 0.238 |
> | MIHC (Ours) | **0.049** | **0.055** | **0.428** | **0.384** | **0.281** | **0.062** | **0.067** | **0.399** | **0.357** | **0.262** |
> | **Grid-based** |
> | CircuitGNN | 0.064 | 0.070 | 0.348 | 0.148 | 0.122 | 0.077 | 0.083 | 0.318 | 0.138 | 0.112 |
> | LHNN | 0.060 | 0.066 | 0.369 | 0.162 | 0.135 | 0.073 | 0.079 | 0.337 | 0.152 | 0.125 |
> | Lay-Net | 0.055 | 0.061 | 0.392 | 0.197 | **0.179** | 0.068 | 0.073 | 0.367 | **0.191** | 0.152 |
> | MIHC (Ours) | **0.054** | **0.059** | **0.415** | **0.202** | 0.175 | **0.067** | **0.072** | **0.388** | 0.182 | **0.168** |
>
>
> [1] Lay-Net: Grafting Netlist Knowledge on Layout-Based Congestion Prediction. TCAD 2025
>
>
> >W3: Additionally there are quite a few components to the model and it does seem a little complex. However, that is often the case in chip design methods. Finally I'd like to see more about training cost and frequency.
>
> **A:**
> We acknowledge that MIHC involves multiple components, but each serves a specific purpose essential for balanced fusion of netlist and layout information and providing crucial interpretability for circuit design validation. As you noted, this complexity is typical in chip design methods given the inherent complexity of circuit data. As shown in Appendix C.8, our experiments on one NVIDIA A100 80GB GPU demonstrate practical performance: small circuits (~30K cells/nets) require ~0.9 seconds per training iteration and 3GB memory, while very large circuits (1.3M cells/nets) take ~5.0 seconds per training iteration and 22GB memory. Based on our experimental setup, the total training time is approximately 1.5 hours for ISPD2015-B, 2.8 hours for ISPD2015-F, and 15.6 hours for CircuitNet-N28. We will include this discussion in the revised version. These measurements confirm that despite the model's sophistication, its computational demands remain manageable for real-world industrial applications.
>
> >Q2: How often do you train a new model and how much compute does it take?
>
> **A:** As shown in Appendix C.8, our experiments on one NVIDIA A100 80GB GPU demonstrate practical performance: small circuits (~30K cells/nets) require ~0.9 seconds per training iteration and 3GB memory, while very large circuits (1.3M cells/nets) take ~5.0 seconds per training iteration and 22GB memory. Based on our experimental setup, the total training time is approximately 1.5 hours for ISPD2015-B, 2.8 hours for ISPD2015-F, and 15.6 hours for CircuitNet-N28. These measurements confirm that despite the model's sophistication, its computational demands remain manageable for real-world industrial applications.

---

> ### Comment · Area_Chair_uRFj · 2025-08-05
>
> Dear Reviewer ibfY,
>
> Please join the discussion to assess whether the authors’ response has adequately addressed your questions and concerns. Thank you very much for your engagement!
>
> Best regards,
> AC

---

> ### Author Response · Authors · 2025-08-07
> **Request for Your Valuable Feedback on Our Rebuttal**
>
> Dear Reviewer ibfY,
>
> Thank you for your thorough review and constructive feedback on our paper about MIHC for congestion prediction in chip design. We deeply appreciate the time and effort you've dedicated to evaluating our work, particularly your insightful concerns about generalization and training costs.
>
> We have addressed your questions regarding **model generalization across different designs**, **the training/test split methodology**, and **computational requirements** in our detailed rebuttal. We particularly provided comprehensive additional experiments with unseen design splits (following Lay-Net's Exp2 setup) to demonstrate genuine generalization capability, along with detailed training cost analysis showing practical computational requirements for real-world deployment.
>
> We would be extremely grateful if you could share your thoughts on our response and let us know if our additional generalization experiments and training cost analysis have adequately addressed your concerns, or if there are any further questions we could clarify.
>
> Thank you again for your time and consideration.
>
> Best regards,
>
> All the Authors

---

> > ### Comment · Reviewer_ibfY · 2025-08-08
> >
> > I appreciate the detailed comments and extra experiments. I am raising my score to a "borderline accept". I think this is a decent paper but I am not 100% convinced of its generalizability.

---

> > > ### Author Response · Authors · 2025-08-09
> > >
> > > Dear Reviewer ibfY,
> > >
> > > Thank you very much for your thoughtful engagement with our rebuttal and for raising your score. We greatly appreciate your acknowledgment of the additional unseen design experiments and training cost analysis we provided.
> > >
> > > We understand your remaining reservations about generalizability, and we agree that this is an important aspect in the field of congestion prediction. Based on our current work, we believe this aspect will naturally lead to fruitful directions for future research.
> > >
> > > Thank you again for your constructive feedback throughout the review process, which has significantly helped improve our work.
> > >
> > > Best regards,
> > >
> > > All the Authors

---

### Decision · Program_Chairs · 2025-09-17

**Decision:**

Accept (poster)

**Comment:**

This paper introduces a multi-view hypergraph neural network that predicts chip congestion by simultaneously processing netlist (topological) and layout (geometric) data. To improve accuracy and trustworthiness, it uses a novel subgraph Information Bottleneck mechanism to identify and focus on critical, congestion-related circuit regions.

The paper's strengths lie in its novel and balanced approach to fusing multi-view circuit data within a unified hypergraph framework and its pioneering use of an Information Bottleneck for interpretable, subgraph-level predictions in this domain. The weaknesses include the added computational complexity from the IB mechanism and concerns about generalization, particularly on unseen circuit designs. In the rebuttal period, the authors provided runtime analysis showing the cost is manageable for industrial applications, which mitigates this concern as a practical barrier. Overall, this paper makes a good contribution, so I recommend acceptance.